# Deep Batch Active Anomaly Detection With Diverse Queries

## Abstract

Selecting informative data points for expert feedback can significantly improve the performance of anomaly detection in various contexts, such as medical diagnostics or fraud detection. In this paper, we determine a set of conditions under which the ranking of anomaly scores generalizes from labeled queries to unlabeled data. Inspired by these conditions, we propose a new querying strategy for batch active anomaly detection that leads to systematic improvements over current approaches. It selects a diverse set of data points for labeling, achieving high data coverage with a limited budget. These labeled data points provide weak supervision to the unsupervised anomaly detection problem. However, correctly identifying anomalies in the contaminated training data requires an estimate of the contamination ratio. We show how this anomaly rate can be estimated from the query set by importance-weighting, removing the associated bias due to the non-uniform sampling procedure. Extensive experiments on image, tabular, and video data sets show that our approach results in state-of-the-art active anomaly detection performance.

## 1 Introduction

Detecting anomalies in data is a fundamental task in machine learning with applications in various domains, from industrial fault detection to medical diagnosis. The main idea is to train a model (such as a neural network) on a data set of "normal" samples to minimize the loss of an auxiliary (e.g., self-supervised) task. Using the loss function to score test data, one hopes to obtain low scores for normal data and high scores for anomalies (Ruff et al., 2021). Oftentimes, the training data is contaminated with unlabeled anomalies, and many approaches either hope that training will be dominated by the normal samples (*inlier priority*, Wang et al. (2019)) or try to detect and exploit anomalies in the training data (e.g., Qiu et al. (2022a)).

In some set-ups, expert feedback is available to check if individual samples are normal or should be considered anomalies. These labels are usually expensive to obtain but are very valuable to guide an anomaly detector during training. For example, in a medical setting, one may ask a medical doctor to confirm whether a given image shows normal or abnormal cellular tissue. Other application areas include detecting network intrusions or machine failures. As expert feedback is typically expensive, it is essential to find effective strategies for querying informative data points.

Previous work on active anomaly detection primarily involves domain-specific applications and/or ad hoc architectures, making it hard to disentangle modeling choices from querying strategies (Trittenbach et al., 2021). This paper aims to disentangle different factors that affect detection accurary. We theoretically analyze generalization performance under various querying strategies and find that diversified sampling systematically improve over existing popular querying strategies, such as querying data based on their predicted anomaly score or around the decision boundaries.

Based on these findings, we propose active latent outlier exposure (ALOE): a state-of-the-art active learning strategy compatible with many unsupervised and self-supervised losses for anomaly detection (Ruff et al., 2021; Qiu et al., 2022a). ALOE draws information from both queried and unqueried parts of the data based on two equally-weighted losses. Its sole hyperparameter—the assumed anomaly rate—can be efficiently estimated based on an importance sampling estimate. We show on a multitude of data sets (images, tabular data, and video) that ALOE leads to a new state of the art.

In summary, our main contributions are as follows:

1. We prove that the ranking of anomaly scores generalizes from labeled queries to unlabeled data under certain conditions that characterize how well the queries cover the data. Based on this theory, we propose a diverse querying strategy for batch active anomaly detection.
2. We propose ALOE, a new active learning framework compatible with a large number of deep anomaly detection losses. It trains on both the labeled queries and the unlabeled data and we characterize in Thm. 1 how the performance on the queried samples generalizes to the unlabeled training data. We also show how all major hyperparameters in ALOE can be eliminated, making the approach easy to use. To this end, we provide an importance-sampling-based estimate for the rate of anomalies in the data.
3. We provide an extensive benchmark for deep active anomaly detection. Our experiments on image, tabular, and video data provide evidence that ALOE with diverse querying, outperforms existing methods significantly. Comprehensive ablations disentangle the benefits of each component.

Our paper is structured as follows. Section 2 discusses related work in deep active anomaly detection. Section 3 introduces our main algorithm. Section 4 discusses experimental result on image, video, and tabular data. Finally, we conclude this work in Section 5.

## 2 RELATED WORK

**Deep Anomaly Detection.** Many recent advances in anomaly detection are in the area of deep learning Ruff et al. (2021). One early strategy was to use autoencoder- (Principi et al., 2017; Zhou and Paffenroth, 2017; Chen and Konukoglu, 2018) or density-based (Schlegl et al., 2017; Deecke et al., 2018) models. Another pioneering stream of research combines one-class classification (Schölkopf et al., 2001) with deep learning for unsupervised (Ruff et al., 2018) and semi-supervised (Ruff et al., 2019) anomaly detection. Many other approaches to deep anomaly detection are self-supervised. They employ a self-supervised loss function to train the detector and score anomalies (Golan and El-Yaniv, 2018; Hendrycks et al., 2019; Bergman and Hoshen, 2020; Qiu et al., 2021; Shenkar and Wolf, 2022; Qiu et al., 2022b). Our work resides in the self-supervised anomaly detection category and can be extended to other data modalities if an appropriate loss is provided.

While all these methods assume that the training data consists of only normal samples, in many practical applications, the training pool may be contaminated with unidentified anomalies (Vilhjálmsson and Nordborg, 2013; Steinhardt et al., 2017). This can be problematic because the detection accuracy typically deteriorates when the contamination ratio increases (Wang et al., 2019). Addressing this, refinement (Zhou and Paffenroth, 2017; Yoon et al., 2021) attempts to cleanse the training pool by removing anomalous samples therein, although they may provide valuable training signals. As a remedy, Qiu et al. (2022a) propose to jointly infer binary labels to each datum (normal vs. anomalous) while updating the model parameters of a deep anomaly detector based on outlier exposure. Our work also makes the contaminated data assumption and employs the training signal of abnormal data.

**Active Anomaly Detection.** Active learning for anomaly detection was pioneered in Pelleg and Moore (2004). Most research in active anomaly detection is on shallow detectors. Many works query samples that are close to the decision boundary of a one-class SVM (Görnitz et al., 2013; Yin et al., 2018) or a density model (Ghasemi et al., 2011). Siddiqui et al. (2018); Das et al. (2016) propose to query the most anomalous instance, while Das et al. (2019) employ a tree-based ensemble to query both anomalous and diverse samples. A recent survey compares various aforementioned query strategies applied to one-class classifiers (Trittenbach et al., 2021).

Recently, deep active anomaly detection has received a lot of attention. Pimentel et al. (2020) query samples with the top anomaly scores for autoencoder-based methods, while Ning et al. (2022) improve the querying by taking diversity into consideration. Tang et al. (2020) use an ensemble of deep anomaly detectors and query the most likely anomalies for each detector separately. Russo et al. (2020) query samples where the model is uncertain about the predictions. Pang et al. (2021) and Zha et al. (2020) propose querying strategies based on reinforcement learning, which requires additional labeled datasets. In contrast, our work does not require a labeled dataset to start active learning. Our work is also not comparable to Pelleg and Moore (2004) and Ghasemi et al. (2011), who fit a density model on the raw input data, which is known to be problematic for high-dimensional data (Nalisnick et al., 2018). Other querying strategies from the papers discussed above are fairly general and can be applied in combination with various backbone models. In this paper we study these methods in the

batch active anomaly detection setting. For fair comparison, we apply their meta-strategies to the same state-of-the-art backbone model that our proposed method uses in our experiments (see details in Tab. 1 and Sec. 4).

**Batch Active Learning.**   Like Hoi et al. (2006a) our work considers the batch active anomaly detection setting. Here it is assumed that labeling can be performed only once. Either because interacting with the expert for labeling is too expensive or because retraining the model with each additional query is too inefficient (Hoi et al., 2006a). The batch setting is widely studied in deep active learning (Sener and Savarese, 2018; Ash et al., 2020; Citovsky et al., 2021; Pinsler et al., 2019; Hoi et al., 2006b; Guo and Schuurmans, 2007) but is still under-explored in active anomaly detection.

## 3 METHODS

A successful method for active anomaly detection will involve a strategy for selecting informative data points to be labeled – these are called the *queries* – and a training strategy that best exploits the labeled (and unlabeled) data. We first review recent deep learning losses for anomaly detection (Sec. 3.1), before proposing a new approach for batch active anomaly detection in Secs. 3.2 to 3.4.

In Sec. 3.2, we analyze querying strategies for active anomaly detection. We determine a set of conditions that guarantee that the ranking in terms of the anomaly score provably generalizes from the labeled queries to the unlabeled data. Based on our theory, we derive a diverse querying strategy. Once the queries are labeled, we can use the union of the queried data and the unlabeled data to train the model. Our proposed loss is presented in Sec. 3.3. This approach has an important hyperparameter $\alpha$ which corresponds to the expected fraction of anomalies in the data. While previous work has to assume that $\alpha$ is known (Qiu et al., 2022a), active learning presents an opportunity to estimate it. The estimate has to account for the fact that the optimal querying strategy derived from our theory in Sec. 3.2 is not i.i.d.. In Sec. 3.4, we provide an estimate of $\alpha$ for any stochastic querying strategy.

### 3.1 BACKGROUND

**Active Anomaly Detection.**   We assume a dataset where each sample is associated with a feature $\boldsymbol{x} \in \mathbb{R}^d$ and a binary anomaly label $y$. The label $y = 0$ stands for normal and $y = 1$ stands for abnormal data. While most samples are normal, the dataset is contaminated with a fraction of $\alpha$ anomalies. The goal is to learn a parametric function $S(\boldsymbol{x})$ that associates every input $\boldsymbol{x}$ with a real-valued anomaly score. A high anomaly score is predictive of $y = 1$ and a low anomaly score is predictive of $y = 0$. To train $S$, one has access to unlabeled data with index set denoted by $\mathcal{U}$ and labeled data with index set denoted by $\mathcal{Q}$. The number of queries corresponds to the labeling budget $|\mathcal{Q}|$. Once $S$ is trained, it can be used to rank new samples according to their anomaly score.

**Supervised Anomaly Detection.**   One possibility is to use only the labeled data to train $S$ in a supervised manner (Görnitz et al., 2013; Hendrycks et al., 2018). It requires two related loss functions with shared parameters $\theta$ parameterizing the function $S$. The first loss $L_0^\theta(\boldsymbol{x}) \equiv L_0(S(\boldsymbol{x}; \theta))$ only involves normal samples, while $L_1^\theta(\boldsymbol{x}) \equiv L_1(S(\boldsymbol{x}; \theta))$ is the loss for the anomalies. Options for $L_0^\theta(\boldsymbol{x})$ include an autoencoder loss, the Deep SVDD loss, or a neural transformation learning objective (Zhou and Paffenroth, 2017; Ruff et al., 2018; Qiu et al., 2021). The loss $L_1^\theta(\boldsymbol{x})$ for abnormal data, is typically chosen to have a complementary effect [1]. With the labeled queries $\mathcal{Q}$, the supervised loss is

$$\mathcal{L}_\mathcal{Q}(\theta) = \frac{1}{|\mathcal{Q}|} \sum_{j \in \mathcal{Q}} \left( y_j L_1^\theta(\boldsymbol{x}_j) + (1 - y_j) L_0^\theta(\boldsymbol{x}_j) \right). \tag{1}$$

**Latent Outlier Exposure.**   Latent outlier exposure (LOE, (Qiu et al., 2022a)) is an unsupervised anomaly detection framework that uses the same loss as Eq. (1) but treats the labels $\boldsymbol{y}$ as latent variables. An EM-style algorithm alternates between optimizing the model parameters $\theta$ and inferring the labels $\boldsymbol{y}$. For this, Qiu et al. (2022a) have to assume that the ranking of data anomaly scores is correct and the contamination ratio $\alpha$ is known.

---

[1] For example, in Deep SVDD (Ruff et al., 2018), $L_0^\theta(\boldsymbol{x})$ is the squared distance between $\boldsymbol{x}$ and a normality center in feature space, and $L_1^\theta(\boldsymbol{x}) = 1/L_0^\theta(\boldsymbol{x})$.

In this work, we propose active latent outlier exposure (ALOE). We next present the querying strategy and when the querying strategy leads to correct ranking of anomaly scores (Sec. 3.2), the ALOE loss (Sec. 3.3), and how the hyperparameter $\alpha$ can be eliminated (Sec. 3.4).

## 3.2 ACTIVE QUERYING FOR ANOMALY DETECTION

The first ingredient of an active anomaly detection method is a querying strategy for selecting informative data points to be labeled. Several strategies have been proposed in the literature. As a simple baseline, data points can be queried at random (Ruff et al., 2019); however, this oftentimes results in very few queried true anomalies. Görnitz et al. (2013) propose to query at the boundary of the normal region, which is the learned hypersphere surface in the feature space. A more general approach is to query data in the $\alpha$-quantile of an anomaly detector, but the problem is that one typically does not know which value of $\alpha$ defines the decision boundary.

An important property of the querying strategy is how well it covers unlabeled data. The quality of a querying strategy is determined by the smallest radius $\delta$ such that all unlabeled points are within distance $\delta$ of one queried sample of the same type. In this paper, we prove that if the queries cover both the normal data and the anomalies well (i.e., if $\delta$ is small), a learned anomaly detector that satisfies certain conditions is guaranteed to generalize correctly to the unlabeled data (The exact statement and its conditions will be provided in Thm. 1). Based on this insight, we propose to use a querying strategy that is better suited for active anomaly detection than previous work.

**Theorem 1.** *Let $\mathcal{Q}_0$ be the index set of datapoints labeled normal and $\mathcal{Q}_1$ the index set of datapoints labeled abnormal. Let $\delta \in \mathbb{R}^+$ be the smallest radius, such that for each unlabeled anomaly $\boldsymbol{u}_a$ and each unlabeled normal datum $\boldsymbol{u}_n$ there exist labeled data points $\boldsymbol{x}_a, a \in \mathcal{Q}_1$ and $\boldsymbol{x}_n, n \in \mathcal{Q}_0$, such that $\boldsymbol{u}_a$ is within the $\delta$-ball of $\boldsymbol{x}_a$ and $\boldsymbol{u}_n$ is within the $\delta$-ball around $\boldsymbol{x}_n$. If a $\lambda_s$-Lipschitz continuous function $S$ ranks the labeled data correctly, with a large enough margin, i.e. $S(\boldsymbol{x}_a) - S(\boldsymbol{x}_n) \geq 2\delta\lambda_s$, then $S$ ranks the unlabeled points correctly too and $S(\boldsymbol{u}_a) \geq S(\boldsymbol{u}_n)$.*

*Proof.* Since $S$ is $\lambda_s$-Lipschitz continuous and $\boldsymbol{u}_a$ and $\boldsymbol{u}_n$ are assumed to be closer than $\delta$ to $\boldsymbol{x}_a$ and $\boldsymbol{x}_n$ respectively, we have $S(\boldsymbol{x}_a) - \delta\lambda_s \leq S(\boldsymbol{u}_a)$ and $-S(\boldsymbol{x}_n) - \delta\lambda_s \leq -S(\boldsymbol{u}_n)$. Adding the inequalities and using the condition $S(\boldsymbol{x}_a) - S(\boldsymbol{x}_n) \geq 2\delta\lambda_s$, yields $0 \leq S(\boldsymbol{x}_a) - S(\boldsymbol{x}_n) - 2\delta\lambda_s \leq S(\boldsymbol{u}_a) - S(\boldsymbol{u}_n)$, which proves $S(\boldsymbol{u}_a) \geq S(\boldsymbol{u}_n)$. □

An implication of this theorem is that a smaller $\delta$ corresponding to a tighter cover of the data leads to better generalization performance. Given a fixed query budget $|\mathcal{Q}|$, random querying puts too much weight on high-density areas of the data space, while other strategies only query locally, e.g., close to the possible decision boundary between normal and abnormal data. Instead, we propose a querying strategy that encourages diversity: k-means++ , the seeding algorithm that initializes diverse cluster centers. It iteratively samples another data point to be added to the query set $\mathcal{Q}$ until the labeling budget is reached. Given the existing queried samples, the probability of drawing a sample from the unlabeled set $\mathcal{U}$ is proportional to its distance to the closest sample in the query set:

$$p_{\text{query}}(\boldsymbol{x}_i) = \text{softmax}\big(h(\boldsymbol{x}_i)/\tau\big) \quad \forall i \in \mathcal{U}, \tag{2}$$

where $\tau$ is a temperature parameter controlling the strength of diversity, $h(\boldsymbol{x}_i) = \min_{\boldsymbol{x}_j \in \mathcal{Q}} d(\boldsymbol{x}_i, \boldsymbol{x}_j)$ is the distance of a sample $\boldsymbol{x}_i$ to the query set $\mathcal{Q}$. For a more meaningful notion of distance, we define the latter in an embedding space as $d(\boldsymbol{x}, \boldsymbol{x}') = \|\phi(\boldsymbol{x}) - \phi(\boldsymbol{x}')\|_2$, where $\phi$ is a neural feature map. We stress that all deep methods considered in this paper have an associated feature map that we can use. In Supp. D.5, we report the time complexity. In Supp. A, we empirically validate that diverse querying leads to smaller $\delta$ than others and is therefore advantageous for active anomaly detection.

## 3.3 ACTIVE LATENT OUTLIER EXPOSURE (ALOE) LOSS

We next consider how to use both labeled and unlabeled samples in training. We propose active latent outlier exposure (ALOE). Its loss combines the unsupervised anomaly detection loss of LOE (Qiu et al., 2022a) for the unlabeled data with the supervised loss (Eq. (1)) for the labeled samples. For all queried data (with index set $\mathcal{Q}$), we assume that ground truth labels $y_i$ are available, while for unqueried data (with index set $\mathcal{U}$), the labels $\tilde{y}_i$ are unknown. Adding both losses together yields

$$\mathcal{L}(\theta, \tilde{\boldsymbol{y}}) = \frac{1}{|\mathcal{Q}|} \sum_{j \in \mathcal{Q}} \big(y_j L_1^\theta(\boldsymbol{x}_j) + (1 - y_j) L_0^\theta(\boldsymbol{x}_j)\big) + \frac{1}{|\mathcal{U}|} \sum_{i \in \mathcal{U}} \big(\tilde{y}_i L_1^\theta(\boldsymbol{x}_i) + (1 - \tilde{y}_i) L_0^\theta(\boldsymbol{x}_i)\big). \tag{3}$$

Similar to Qiu et al. (2022a), optimizing this loss involves a block coordinate ascent scheme that alternates between inferring the unknown labels and taking gradient steps to minimize Eq. (3) with the inferred labels. In each iteration, the pseudo labels $\tilde{y}_i$ for $i \in \mathcal{U}$ are obtained by minimizing Eq. (3) subject to a constraint of $\sum_{i \in \mathcal{Q}} y_i + \sum_{i \in \mathcal{U}} \tilde{y}_i = \alpha N$. The constraint ensures that the inferred anomaly labels respect a certain contamination ratio $\alpha$. In practice, this constrained optimization problem is solved by using the current anomaly score function $S$ to rank the unlabeled samples and assign the top $\alpha$-quantile of the associated labels $y_i$ to the value 1, and the remaining to the value 0. Thm. 1 applies in ranking the unlabeled samples by the anomaly score function $S$.

In theory, $\alpha$ could be treated as a hyperparameter, but eliminating hyperparameters is important in anomaly detection. In many practical applications of anomaly detection, there is no labeled data that can be used for validation. While Qiu et al. (2022a) have to assume that the contamination ratio is given, active learning provides an opportunity to estimate $\alpha$. In Sec. 3.4, we develop an importance-sampling based approach to estimate the contamination ratio from the labeled data. Estimating this ratio can be beneficial for many anomaly detection algorithms, including OC-SVM (Schölkopf et al., 2001), kNN (Ramaswamy et al., 2000), Robust PCA/Auto-encoder (Zhou and Paffenroth, 2017), and Soft-boundary deep SVDD (Ruff et al., 2018). When working with contaminated data, these algorithms require a decent estimate of the contamination ratio for good performance.

Another noteworthy aspect of the ALOE loss is that it weighs the *averaged* losses equally to each other. While equal weighting can not be justified from the first principles, we expect our queried samples to be informative, and yet we do not want either loss component to dominate the learning task (assuming that we always query only a small percentage of the data). In Supp. E.5, we empirically show that equal weighting yields the best results among a large range of various weights. This provides more weight to every queried data point than to an unqueried one, assuring these goals. Our equal weighting scheme is also practical because it avoids a hyperparameter.

### 3.4 CONTAMINATION RATIO ESTIMATION.

To eliminate a critical hyperparameter in our approach, we estimate the *contamination ratio* $\alpha$, i.e., the fraction of anomalies in the dataset. Under a few assumptions, we show that this parameter can be estimated despite of the result of Thm. 1, which requires us to use a biased querying strategy.

We consider the contamination ratio $\alpha = \mathbb{E}_{p(\boldsymbol{x})}[\mathbb{1}_a(\boldsymbol{x})]$, where $p(\boldsymbol{x})$ is the data distribution and $\mathbb{1}_a(\boldsymbol{x}) \equiv \mathbb{1}(y(\boldsymbol{x}) = 1)$ indicates whether or not the sample is abnormal. Estimating this quantity would be trivial given an unlimited budget of queries. However, since active learning setups involve limited budgets, we aim to estimate this quantity only based on the queried samples $\mathcal{Q}$ used to train the model.

Since the queried data points are not independently sampled, we cannot straightforwardly estimate $\alpha$ based on the empirical frequency of anomalies in $\mathcal{Q}$. More precisely, our querying procedure results in a chain of indices $\mathcal{Q} = \{i_1, i_2, ..., i_{|\mathcal{Q}|}\}$, where $i_1 \sim \text{Unif}(1:N)$, and each conditional distribution $i_k | i_{<k}$ is defined by Eq. (2). We will show as follows that this sampling bias can be compensated using importance sampling.

As follows, we will state two assumptions about our querying strategy, followed by a theorem for calculating $\alpha$. Justifications for the two assumptions will be provided below.

**Assumption 1.** *The anomaly scores $\{S(\boldsymbol{x}_i) : i \in Q\}$ in a query set $Q$ are approximately independently distributed, i.e., their off-diagonal correlations are negligible.*

**Assumption 2.** *Let $q(\boldsymbol{x}_i)$ denote the marginal distribution of a queried sample under the querying scheme $Q$, and let $p(\boldsymbol{x})$ denote the data distribution. Then, the anomaly scores $S(\boldsymbol{x})$ are a sufficient statistic of both $p(\boldsymbol{x})$ and $q(\boldsymbol{x}_i)$. That is, there exist functions $f(\cdot)$ and $g(\cdot)$ such that $p(\boldsymbol{x}) = f(S(\boldsymbol{x}))$ and $q(\boldsymbol{x}_i) = g(S(\boldsymbol{x}_i))$ for all $\boldsymbol{x}$ and $\boldsymbol{x}_i$, with negligible error.*

Both Assumptions 1 and 2 are only approximations to reality. We will provide strong evidence below that they are well-justified working hypotheses. The following theorem is a consequence of them:

**Theorem 2.** *Given Assumptions 1 and 2, the following statement applies: for a random query $Q$, the following importance sampling estimator provides an unbiased estimate of the contamination ratio:*

$$\hat{\alpha} = \frac{1}{|\mathcal{Q}|} \sum_{i=1}^{|\mathcal{Q}|} \frac{f(S(\boldsymbol{x}_i))}{g(S(\boldsymbol{x}_i))} \mathbb{1}_a(\boldsymbol{x}_i). \tag{4}$$

The proof is in Supp. B. In practice, we learn $f$ and $g$ using a kernel density estimate in the one-dimensional space of anomaly scores. We set the bandwidth to the average spacing of scores. In conclusion, Thm. 2 allows us to estimate the importance ratio based on a non-iid query set $Q$.

**Discussion.** We empirically verified the fact that Thm. 2 results in reliable estimates for varying contamination ratios in Supp. B.4. Since Assumptions 1 and 2 seem strong, we discuss their justifications and empirical validity next.

We tested Assumption 1 empirically and found negligible off-diagonal correlations among the anomaly scores in the query set. In Supp. B.2, we show that the absolute off-diagonal coefficient values are significantly smaller than one on CIFAR-10. We can understand this based on a dimensionality argument. Since we sample repulsively in a high-dimensional space but project the data onto the one-dimensional space of scores, the resulting correlations will vanish upon the projection step.

We tested Assumption 2 empirically in Supp. B.3. We show that there is a monotonic relationship between $S(\boldsymbol{x})$ and $\log p(\boldsymbol{x})$ (or $\log q(\boldsymbol{x})$). We validated both on 2D synthetic data from Gaussian distribution and on CIFAR-10. Scatter plots in Supp. B show that $S(\boldsymbol{x})$ is predictive of both $p(\boldsymbol{x})$ and $q(\boldsymbol{x})$ and thus $S(\boldsymbol{x})$ is a sufficient statistic of $p(\boldsymbol{x})$ and $q(\boldsymbol{x})$.

## 4    EXPERIMENTS

We study batch active anomaly detection on standard image benchmarks, medical images, tabular data, and surveillance videos. Our extensive empirical study establishes how our proposed method compares to eight recent active anomaly detection methods we implemented as baselines. We first describe the baselines and their implementations (Tab. 1) and then the experiments on images (Sec. 4.1), tabular data (Sec. 4.2), videos (Sec. 4.3) and finally additional experiments (Sec. 4.4).

**Baselines.**    Most existing baselines apply their proposed querying and training strategies to shallow anomaly detection methods or sub-optimal deep models (e.g., autoencoders (Zhou and Paffenroth, 2017)). In recent years, these approaches have consistently been outperformed by self-supervised anomaly detection methods (Hendrycks et al., 2019). For a fair comparison of active anomaly detection frameworks, we endow all baselines with the same self-supervised backbone models also used in our method. By default we use neural transformation learning (NTL) (Qiu et al., 2021) as the backbone model, which was identified as state-of-the-art in a recent independent comparison of 13 models (Alvarez et al., 2022). Results with other backbone models are shown in Supp. E.1.

The baselines are summarized in Tab. 1 and detailed in Supp. C. They differ in their querying strategies (col. 3) and training strategies (col. 4 & 5): the unlabeled data is either ignored or modeled with a one-class objective. Most baselines incorporate the labeled data by a supervised loss (Eq. (1)). As an exception, Ning et al. (2022) remove all queried anomalies and then train a weighted one-class objective on the remaining data. All baselines weigh the unsupervised and supervised losses equally.

The baselines also differ in their querying strategies, summarized below:

- **margin query** selects samples deterministically that are the closest to the boundary of the normality region. We provide the method with the true contamination ratio to help it choose an ideal boundary.
- **margin diverse query** combines margin query with neighborhood-based diversification. It tends to select samples that are not in the $k$-nearest neighbors of the queried set. Thus samples are ensured to sit at different positions of the normality boundary.
- **most positive query** always selects the top-ranked samples ordered by their anomaly scores.
- **positive diverse query** combines querying according to anomaly scores with distance-based diversification. The selection criterion is an equally weighted combination of the two aspects: anomaly score and the minimum Euclidean distance to all previously queried samples.
- **random query** draws samples uniformly among the training set.
- **positive random query** samples uniformly among the top $50\%$ data ranked by anomaly scores.

**Implementation Details.**    In all experiments, we use a NTL (Qiu et al., 2021) backbone model for all methods. Experiments with other backbone models are shown in Supp. E.1. On images and videos, NTL is built upon the penultimate layer output of a frozen ResNet-152 pre-trained on ImageNet. NTL is trained for one epoch, after which all queries are labeled. The number of queries $|\mathcal{Q}|$ depends

Table 1: A summary of all compared experimental methods' query strategy and training strategy irrespective of their backbone models.

| Name | Reference | Querying Strategy | Loss (labeled) | Loss (unlabeled) |
|------|-----------|-------------------|----------------|------------------|
| Mar | Görnitz et al. (2013) | margin query | superv. (Eq. (1)) | one class |
| Hybr1 | Görnitz et al. (2013) | margin diverse query | superv. (Eq. (1)) | one class |
| Pos1 | Pimentel et al. (2020) | most positive query | superv. (Eq. (1)) | none |
| Pos2 | Barnabé-Lortie et al. (2015) | most positive query | superv. (Eq. (1)) | one class |
| Rand1 | Ruff et al. (2019) | random query | superv. (Eq. (1)) | one class |
| Rand2 | Trittenbach et al. (2021) | positive random query | superv. (Eq. (1)) | one class |
| Hybr2 | Das et al. (2019) | positive diverse query | superv. (Eq. (1)) | none |
| Hybr3 | Ning et al. (2022) | positive diverse query | refinement | weighted one class |
| ALOE | [ours] | diverse (Eq. (2)) | active latent outlier exposure loss (Eq. (3)) | |

on the experiment and all $|\mathcal{Q}|$ queries are collected at once. The contamination ratio $\alpha$ in ALOE is estimated immediately after the querying step and then fixed for the remaining training process. We follow Qiu et al. (2022a) and set $\tilde{y}_i = 0.5$ for inferred anomalies. This accounts for the uncertainty of whether the sample truly is an anomaly. More details are provided in Supp. D.

## 4.1 EXPERIMENTS ON IMAGE DATA

In this section, we study ALOE on standard image benchmarks to establish how it compares to eight well-known active anomaly detection baselines. Active learning plays an important role in medical domains where expert labeling is expensive. Hence, we also study nine medical datasets from Yang et al. (2021). We describe the datasets, the evaluation protocol, and finally the results of our study.

**Image Benchmarks.** We experiment with two popular image benchmarks: CIFAR-10 and Fashion-MNIST. These have been widely used in previous papers on deep anomaly detection (Ruff et al., 2018; Golan and El-Yaniv, 2018; Hendrycks et al., 2019; Bergman and Hoshen, 2020).

**Medical Images.** Since medical imaging is an important practical application of active anomaly detection, we also study ALOE on medical images. The datasets we consider cover different data modalities (e.g., X-ray, CT, electron microscope) and their characteristic image features can be very different from natural images. Our empirical study includes all 2D image datasets presented in Yang et al. (2021) that have more than 500 samples in each class, including Blood, OrganA, OrganC, OrganS, OCT, Pathology, Pneumonia, and Tissue. We also include Dermatoscope but restricted to the classes with more than 500 training samples.

**Evaluation Protocol.** We follow the community standard known as the "one-vs.-rest" protocol to turn these classification datasets into a test-bed for anomaly detection (Ruff et al., 2018; Golan and El-Yaniv, 2018; Hendrycks et al., 2019; Bergman and Hoshen, 2020). While respecting the original train and test split of these datasets, the protocol iterates over the classes and treats each class in turn as normal. Random samples from the other classes are used to contaminate the data. The training set is then a mixture of unlabeled normal and abnormal samples with a contamination ratio of 10% (Wang et al., 2019; Qiu et al., 2022a). This protocol can simulate a "human expert" to provide labels for the queried samples because the datasets provide ground-truth class labels. The models are trained on the training data and evaluated on the test data. The reported results (in terms of AUC %) for each dataset are averaged over the number of experiments (i.e., classes) and over five independent runs.

**Results.** We report the evaluation results of our method (ALOE) and the eight baselines on all eleven image datasets in Tab. 2. All methods have a query budget of 20 samples. On all datasets, our proposed method ALOE achieves the best performance and significantly outperforms all baselines by 6 percentage points on average, reaching the state-of-the-art results in active anomaly detection.

In addition, we also study detection performance as the query budget increases from 20 to 160. The results are plotted in Fig. 1. The results show that, with a small budget of 20 samples, ALOE (by querying diverse and informative samples) makes better usage of the labeling information than the other baselines and thus leads to better performance by a large margin. As more samples are allowed

Table 2: AUC (%) with standard deviation for anomaly detection on 11 image datasets when the query budget $|\mathcal{Q}| = 20$. ALOE outperforms all baselines by a large margin by querying diverse and informative samples. For all experiments, we set the contamination ratio as $10\%$.

| | Mar | Hybr1 | Pos1 | Pos2 | Rand1 | Rand2 | Hybr2 | Hybr3 | ALOE |
|---|---|---|---|---|---|---|---|---|---|
| **CIFAR10** | 92.4±0.7 | 92.0±0.7 | 93.4±0.5 | 92.1±0.7 | 89.2±3.2 | 91.4±1.0 | 85.1±2.2 | 71.8±7.4 | **96.3±0.3** |
| **FMNIST** | 93.1±0.4 | 92.6±0.4 | 92.2±0.6 | 89.3±1.0 | 84.0±3.8 | 90.6±1.1 | 88.7±1.4 | 82.6±4.3 | **94.8±0.6** |
| **Blood** | 68.6±1.8 | 69.1±1.3 | 69.6±1.8 | 72.2±4.9 | 70.6±1.6 | 69.2±1.7 | 72.2±2.7 | 58.3±5.2 | **80.5±0.5** |
| **OrganA** | 86.4±1.3 | 87.4±0.7 | 81.7±2.9 | 81.8±2.1 | 82.9±0.6 | 86.5±0.7 | 88.6±1.5 | 68.8±3.1 | **90.7±0.7** |
| **OrganC** | 86.5±0.9 | 87.0±0.7 | 84.6±1.9 | 79.6±2.0 | 85.5±0.9 | 86.4±0.8 | 84.8±1.2 | 68.9±3.0 | **89.7±0.7** |
| **OrganS** | 83.5±1.1 | 84.1±0.4 | 83.2±1.3 | 78.6±1.0 | 82.2±1.4 | 83.8±0.4 | 82.3±0.7 | 66.9±4.3 | **87.4±0.8** |
| **OCT** | 64.4±3.7 | 63.3±1.8 | 63.8±4.4 | 63.0±4.0 | 59.7±1.9 | 62.1±4.3 | 63.0±7.6 | 56.2±4.5 | **68.5±3.4** |
| **Path** | 82.7±2.4 | 86.0±1.1 | 77.5±2.0 | 80.2±3.5 | 83.2±1.6 | 83.9±2.9 | 86.1±2.0 | 75.1±4.2 | **88.1±1.1** |
| **Pneumonia** | 72.1±7.0 | 75.1±5.3 | 75.5±8.8 | 83.6±6.1 | 68.1±5.9 | 76.0±8.0 | 88.4±3.3 | 63.4±17.7 | **91.2±1.4** |
| **Tissue** | 60.2±1.5 | 61.3±1.7 | 65.8±1.7 | 63.5±2.0 | 59.9±1.7 | 59.5±1.3 | 62.1±1.7 | 50.8±1.6 | **66.4±1.4** |
| **Derma** | 62.6±3.8 | 63.1±4.7 | 66.6±2.3 | 66.4±4.3 | 64.5±4.8 | 68.3±2.1 | 57.2±13.3 | 48.0±13.6 | **73.5±2.5** |
| **Average** | 77.5 | 78.3 | 77.3 | 77.6 | 75.4 | 78.0 | 78.0 | 64.6 | **84.3** |

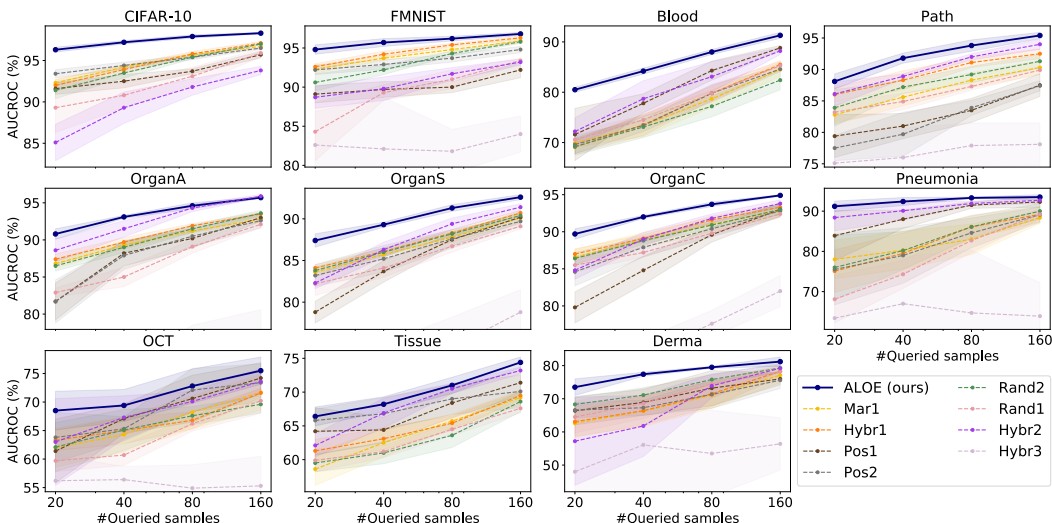

Figure 1: Running AUCs (%) with different query budgets. Models are evaluated at $20, 40, 80, 160$ queries. ALOE performs the best among the compared methods on all query budgets.

to be queried, the performance of almost all methods increases but even for $|\mathcal{Q}| = 160$ queries when the added benefit from adding more queries starts to saturate, ALOE still outperforms the baselines.

## 4.2 Experiments on Tabular Data

Many practical use cases of anomaly detection (e.g., in health care or cyber security) are concerned with tabular data. For this reason, we study ALOE on several tabular datasets from various domains. We find that it outperforms existing baselines, even with as few as 10 queries. We also confirmed the fact that our deep models are competitive with classical methods for tabular data in Supp. E.7.

**Tabular Datasets.** Our study includes the four multi-dimensional tabular datasets from the ODDS repository which have an outlier ratio of at least $30\%$. This is necessary to ensure that there are enough anomalies available to remove from the test set and add to the clean training set (which is randomly sub-sampled to half its size) to achieve a contamination ratio of $10\%$. The datasets are BreastW, Ionosphere, Pima, and Satellite. As in the image experiments, there is one round of querying, in which 10 samples are labeled. For each dataset, we report the averaged F1-score (%) with standard deviations over five runs with random train-test splits and random initialization.

**Results.** Our proposed method ALOE performs best on 3 of 4 datasets and outperforms all baselines by 3.2 percentage points on average, reaching state-of-the-art results in active anomaly detection.

Table 3: F1-score (%) with standard deviation for anomaly detection on tabular data when the query budget $|\mathcal{Q}| = 10$. ALOE performs the best on 3 of 4 datasets. We set the contamination ratio as 10%.

|  | Mar | Hybr1 | Pos1 | Pos2 | Rand1 | Rand2 | Hybr2 | Hybr3 | ALOE |
|---|---|---|---|---|---|---|---|---|---|
| **BreastW** | 81.6±0.7 | 83.3±2.0 | 58.6±7.7 | 81.3±0.8 | 87.1±1.0 | 82.9±1.1 | 55.0±6.0 | 79.6±4.9 | **93.9±0.5** |
| **Ionosphere** | 91.9±0.3 | **92.3±0.5** | 56.1±6.2 | 91.1±0.8 | 91.1±0.3 | 91.9±0.6 | 64.0±4.6 | 88.2±0.9 | 91.8±1.1 |
| **Pima** | 50.1±1.3 | 49.2±1.9 | 48.5±0.4 | 52.4±0.8 | 53.6±1.1 | 51.9±2.0 | 53.8±4.0 | 48.4±0.7 | **55.5±1.2** |
| **Satellite** | 64.2±1.2 | 66.2±1.7 | 57.0±3.0 | 56.7±3.2 | 67.7±1.2 | 66.6±0.8 | 48.6±6.9 | 56.9±7.0 | **71.1±1.7** |
| **Average** | 72.0 | 72.8 | 55.1 | 70.4 | 74.9 | 73.3 | 55.4 | 68.3 | **78.1** |

Diverse querying best utilizes the limited query budget to label the diverse and informative data points, yielding a consistent improvement over existing baselines also on tabular data.

### 4.3 EXPERIMENTS ON VIDEO DATA

Detecting unusual objects in surveillance videos is an important application area for anomaly detection. Due to the large variability in abnormal objects and suspicious behavior in surveillance videos, expert feedback is very valuable to train an anomaly detector (we use NTL as the backbone model) in an active manner. We study ALOE on a public surveillance video dataset (UCSD Peds1). The goal is to accurately detect abnormal video frames that contain non-pedestrian objects with a limited query budget.

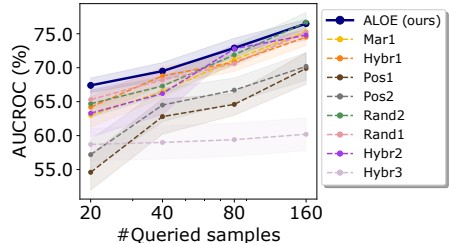

Figure 2: Results on video anomaly detection under varying query budgets. ALOE results in the most accurate active anomaly detector.

We make the same independent frames assumption as Pang et al. (2020) and mix the original training and testing video frames together. As in Pang et al. (2020), we sub-sample 6800 normal frames and 2914 abnormal frames for training such that the anomaly ratio is 0.3, which corresponds to the original ratio in this dataset. The other samples are used for testing. Before running any of the methods, a ResNet pretrained on ImageNet is used to obtain a fixed feature vector for each frame. We vary the query budget from $|\mathcal{Q}| = 20$ to $|\mathcal{Q}| = 160$ and compare ALOE to all baselines. All experiments are repeated five times with different random initializations. Results in terms of average AUC and standard error over the five runs are reported in Fig. 2. ALOE consistently outperforms all baselines, especially for smaller querying budgets.

### 4.4 ADDITIONAL EXPERIMENTS

In Supp. E, we provide several additional experiments and ablations demonstrating ALOE's strong performance and justifying modeling choices. In summary, we address the following aspects:

- **Backbone Models.** Tab. 6 shows that ALOE is the best performing active learning framework also for the backbone models MHRot (Hendrycks et al., 2019) and Deep SVDD (Ruff et al., 2018).
- **ALOE Loss Components.** Tab. 7 shows how different loss components affect performance.
- **Performance under $\hat{\alpha}$.** Fig. 9 shows that ALOE with either true or estimated ratio performs similar given all query budgets.
- **Querying Strategies.** Fig. 8 further confirms the theory of Thm. 1: diverse querying significantly outperforms the other querying strategies in the supervised setting.

## 5 CONCLUSION

We introduced Active Latent Outlier Exposure: a batch active learning approach for deep anomaly detection. Inspired by a set of conditions that guarantee the generalization of anomaly score rankings from queried data to unqueried data, we proposed to use a diversified querying strategy and a combination of two losses for queried and unqueried samples. Based on a simple heuristic for weighting the losses relative to each other and by estimating the unknown contamination rate from queried samples, we were able to make our approach free of its most important hyperparameters. We showed on a variety of data sets from different domains that the approach results in a new state of the art in batch active anomaly detection.

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

## A  THEOREM 1

In Thm. 1, we considered using the fixed-radius neighborhood ($\delta$-ball) of the queried data as the cover of the whole dataset, and mentioned diverse querying has a smaller radius than other querying strategies. In this section, we will empirically verify this fact and further illustrate diverse querying leads to good ranking of un-queried data (see also Fig. 8 on test data).

As defined in Thm. 1, the radius is the smallest distance that is required for any un-queried sample to be covered by the neighborhood of a queried sample of the same type. Mathematically, we compute the radius as

$$\delta = \max_{i \in \mathcal{U}} \min_{j \in \mathcal{Q}, y_i = y_j} d(\boldsymbol{x}_i, \boldsymbol{x}_j), \tag{5}$$

where we adopt the euclidean distance in the feature space for a meaningful metric $d$. We apply NTL on the first class of CIFAR-10 and F-MNIST dataset. We make queries with different budgets, after which we compute $\delta$ by Eq. (5). We repeat this procedure for 100 times and report the mean and standard deviation in Fig. 3. We compared four querying strategies: diverse queries (k-means++ ),

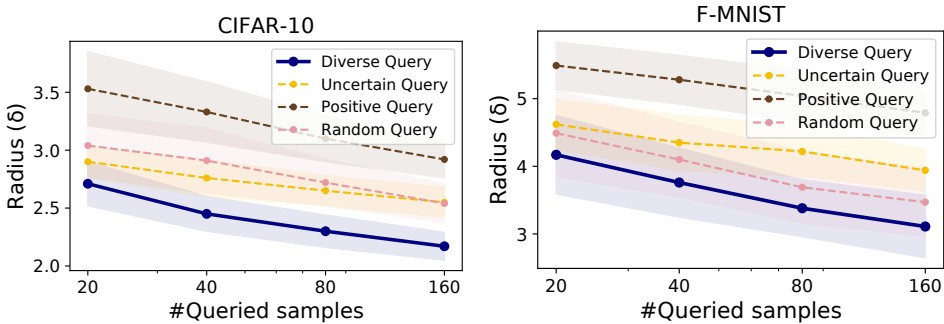

Figure 3: Cover radius $\delta$ (Eq. (5)) resulted from different querying strategies on the first class of CIFAR-10 and F-MNIST. Diverse queries systematically have smaller cover radius than other querying strategies.

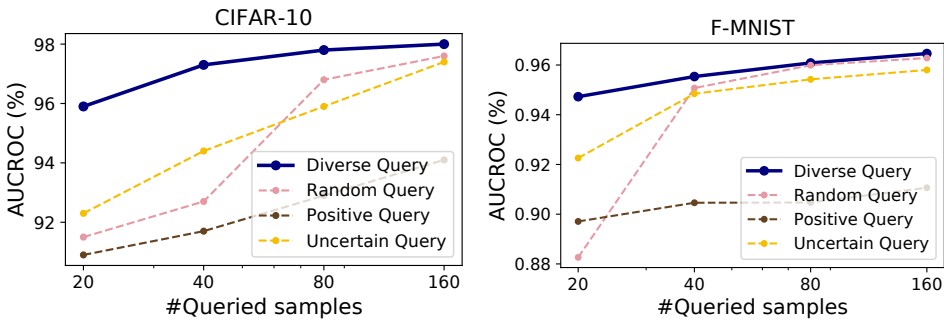

Figure 4: Ranking performance of unlabeled data. AUC of unqueried data is evaluated using the fitted anomaly detector on the queried data. Our proposed diverse querying (k-means++) provides better ranking of the unlabeled data.

uncertain queries (Mar), positive queries (Pos1), and random queries (Rand1). It shows that diverse queries significantly lead the smallest radius $\delta$ among the compared strategies on all querying budgets.

Next, we provide an empirical, overall justification of Thm. 1 (see also Fig. 8 on test data). An implication of Thm. 1 is that, assuming anomaly scores are fixed, a smaller $\delta$ will satisfy the large anomaly score margin $(S(\boldsymbol{x}_a) - S(\boldsymbol{x}_n))$ more easily, hence it is easier for $S$ to correctly rank the remaining unlabeled points. To justify this implication, we need a metric of ranking. AUC satisfies this requirement as it is alternatively defined as (Mohri et al., 2018, 10.5.2)(Cortes and Mohri, 2003)

$$\text{AUC} = \frac{1}{|\mathcal{U}_0| + |\mathcal{U}_1|} \sum_{n \in \mathcal{U}_0, a \in \mathcal{U}_1} \mathbb{1}(S(\boldsymbol{u}_a) > S(\boldsymbol{u}_n)) \approx P_{n \in \mathcal{U}_0, a \in \mathcal{U}_1}(S(\boldsymbol{u}_a) > S(\boldsymbol{u}_n))$$

which measures the probability of ranking unlabeled samples $\boldsymbol{u}_a$ higher than $\boldsymbol{u}_n$ in terms of their scores. $\mathcal{U} = \mathcal{U}_0 \bigcup \mathcal{U}_1$ is the un-queried data indices and $\mathcal{U}_0$ and $\mathcal{U}_1$ are disjoint un-queried normal and abnormal data sets respectively. $\boldsymbol{u}_a$ and $\boldsymbol{u}_n$ are instances of each kind. We conducted experiments on CIFAR-10 and F-MNIST, where we trained an anomaly detector (NTL) on the queried data for 30 epochs and then compute the AUC on the remaining un-queried data. The results of four querying straties are reported in Fig. 4, which shows that our proposed diverse querying strategy generalizes the anomaly score ranking the best to the unqueried data among the compared strategies, testifying our analysis in the main paper. A consequence is that diverse querying can provide accurate assignments of the latent anomaly labels, which will further help learn a high-quality of anomaly detector through the unsupervised loss term in Eq. (3).

# B    THEOREM 2

In this section, we will empirically justify the assumptions we made in Sec. 3.4 that are used to build an unbiased estimator of the anomaly ratio $\alpha$ (Eq. (4)). We will also demonstrate the robustness of the estimation under varying $\alpha$.

## B.1    PROOF

*Proof.* Let A1 and A2 denote Assumptions 1 and 2, respectively. Furthermore, let $q(\boldsymbol{x}_1, ..., \boldsymbol{x}_{|Q|})$ denote the query distribution, $q(\boldsymbol{x}_i)$ its marginals, and $|Q|$ the number of queried samples. Then,

$$
\mathbb{E}[\hat{\alpha}] = \mathbb{E}_{q(\boldsymbol{x}_1, \cdots, \boldsymbol{x}_{|Q|})} \left[ \frac{1}{|Q|} \sum_{i=1}^{|Q|} \frac{f(S(\boldsymbol{x}_i))}{g(S(\boldsymbol{x}_i))} \mathbb{1}_a(\boldsymbol{x}_i) \right] \overset{A1}{=} \mathbb{E}_{q(\boldsymbol{x}_1) \cdots q(\boldsymbol{x}_{|Q|})} \left[ \frac{1}{|Q|} \sum_{i=1}^{|Q|} \frac{f(S(\boldsymbol{x}_i))}{g(S(\boldsymbol{x}_i))} \mathbb{1}_a(\boldsymbol{x}_i) \right]
$$

$$
= \frac{1}{|Q|} \sum_{i=1}^{|Q|} \mathbb{E}_{q(\boldsymbol{x}_i)} \left[ \frac{f(S(\boldsymbol{x}_i)) \mathbb{1}_a(\boldsymbol{x}_i)}{g(S(\boldsymbol{x}_i))} \right] \overset{A2}{=} \frac{1}{|Q|} \sum_{i=1}^{|Q|} \mathbb{E}_{q(\boldsymbol{x}_i)} \left[ \frac{p(\boldsymbol{x}_i) \mathbb{1}_a(\boldsymbol{x}_i)}{q(\boldsymbol{x}_i)} \right] = \alpha. \qquad \square
$$

## B.2    ASSUMPTION 1

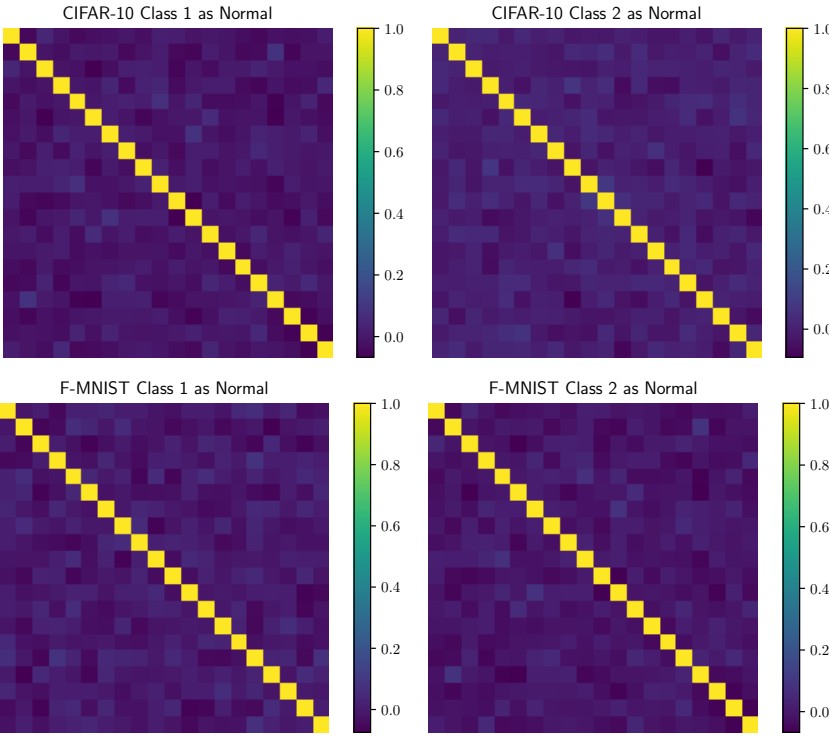

Figure 5: Anomaly score correlation matrix $\langle S(\boldsymbol{x}_i), S(\boldsymbol{x}_j) \rangle$, where $\boldsymbol{x}_i$ and $\boldsymbol{x}_j$ are jointly sampled in the same query set. The result indicates that anomaly scores can be considered as approximately independent random variables.

We verify Assumption 1 by showing the correlation matrix in Fig. 5, where we jointly queried 20 points with diversified querying strategy and repeated 1000 times on two classes of CIFAR-10 and F-MNIST. Then the correlation between each pair of points are computed and placed in the off-diagonal entries. For each matrix, we show the average, maximum, and minimum of the off-diagonal terms

- CIFAR-10 Class 1: -0.001, 0.103, -0.086
- CIFAR-10 Class 2: -0.001, 0.085, -0.094

- F-MNIST Class 1: -0.001, 0.081, -0.075
- F-MNIST Class 2: -0.005, 0.087, -0.067

Which shows the correlations $\langle S(\boldsymbol{x}_i), S(\boldsymbol{x}_j) \rangle$ are negligible, and the anomaly scores can be considered approximately independent random variables.

### B.3  ASSUMPTION 2

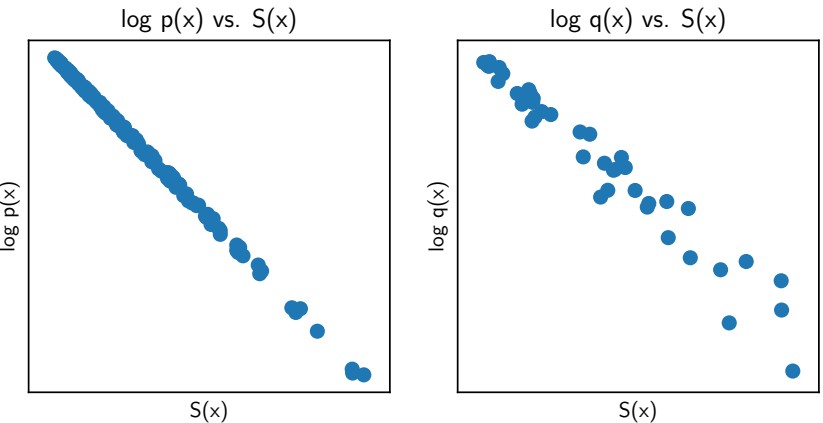

Figure 6: The scatter plot on the left shows a perfect linear relationship between the anomaly score $S(x)$ and $\log p(x)$, and the scatter plot on the right shows a monotonic relationship between $S(x)$ and $\log q(x)$. The results indicate that $S(x)$ is predictive of both $p(x)$ and $q(x)$.

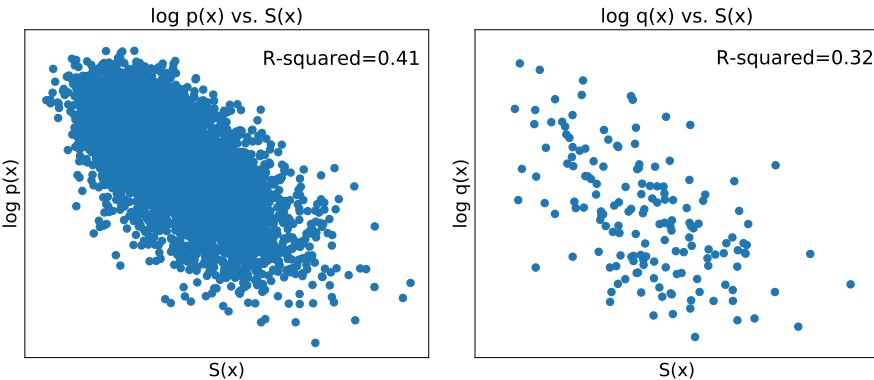

Figure 7: The scatter plot on the left shows a monotonic relationship between the anomaly score $S(x)$ and $\log p(x)$ with a R-squared value of $0.41$, and the scatter plot on the right shows a monotonic relationship between $S(x)$ and $\log q(x)$ with a R-squared value of $0.32$. The results indicate that $S(x)$ is predictive of both $p(x)$ and $q(x)$.

We verify Assumption 2 by checking the correlation between the anomaly score $S(\boldsymbol{x})$ and the data density on one 2D Gaussian-distributed synthetic dataset and one real-world dataset – CIFAR-10. We denote the marginal data distribution by $p(\boldsymbol{x})$, and denote the marginal distribution of a queried sample by $q(\boldsymbol{x})$. For synthetic data, the density $p(\boldsymbol{x})$ is known, and we model $q(\boldsymbol{x})$ with a kernel density estimator. For CIFAR-10, both data densities $p(\boldsymbol{x})$ and $q(\boldsymbol{x})$ are modeled by Masked Autoregressive Flow (Papamakarios et al., 2017). For the choice of the anomaly score function $S$, we learn a deep SVDD (Ruff et al., 2018) for the synthetic data and a NTL for CIFAR-10 with protocols discussed in Sec. 4.1.

Fig. 6 and Fig. 7 plot $S(\boldsymbol{x})$ against $\log p(\boldsymbol{x})$ and $\log q(\boldsymbol{x})$ respectively. In Fig. 7, we observe that there is a monotonic relationship between $S(\boldsymbol{x})$ and $\log p(\boldsymbol{x})$ or $\log q(\boldsymbol{x})$ with a $R^2$ value of $0.41$ and

0.32 respectively. In Fig. 6, we observe a perfect linear relationship between $S(\boldsymbol{x})$ and $\log p(\boldsymbol{x})$, and a monotonic relationship between $S(\boldsymbol{x})$ and $\log q(\boldsymbol{x})$. These plots empirically show that the anomaly score $S(\boldsymbol{x})$ is predictive and a sufficient statistic of both $p(\boldsymbol{x})$ and $q(\boldsymbol{x})$.

### B.4 CONTAMINATION RATIO ESTIMATION

Table 4: Estimated contamination ratios on CIFAR-10 and F-MNIST when $|\mathcal{Q}| = 40$ and the backbone model is NTL. The first row shows the true contamination ratio ranging from 1% to 15%. The estimations are repeated 50 times.

|  | 1% | 5% | 10% | 15% |
|---|---|---|---|---|
| CIFAR-10 | $0.5\% \pm 1.2\%$ | $6.0\% \pm 3.3\%$ | $12.0\% \pm 4.4\%$ | $15.3\% \pm 4.5\%$ |
| F-MNIST | $1.0\% \pm 1.5\%$ | $3.8\% \pm 2.3\%$ | $8.7\% \pm 4.1\%$ | $12.8\% \pm 5.3\%$ |

We estimate the contamination ratio by Eq. (4) under varying true ratios. This part shows the estimated contamination ratio when the query budget is $|\mathcal{Q}| = 40$. The estimations from the backbone model NTL is shown in Tab. 4. The first row contains the ground truth contamination rate, and the second and third row indicate the inferred values for two datasets, using our approach. Most estimates are withing the error bars and hence accurate. The estimation error is acceptable as confirmed by the sensitivity study in (Qiu et al., 2022a) which concludes that the LOE approach still works well if the anomaly ratio is mis-specified within 5 percentage points.

## C  BASELINES DETAILS

In this section, we describe the details of the baselines in Tab. 1 in the main paper. For each baseline method, we explain their query strategies and post-query training strategies we implement in our experiment. Please also refer to our codebase for practical implementation details.

- **Rand1.** This strategy used by Ruff et al. (2019) selects queries by sampling uniformly without replacement across the training set, resulting in the queried index set $\mathcal{Q} = \{i_q \sim \text{Unif}(1, \cdots, N) | 1 \leq q \leq |\mathcal{Q}|\}$. After the querying, models are trained with a supervised loss function based on outlier exposure on the labeled data and with a one-class classification loss function on the unlabeled data,

$$L_{\text{Rand1}}(\theta) = \frac{1}{|\mathcal{Q}|} \sum_{j \in \mathcal{Q}} \left( y_j L_1^\theta(\boldsymbol{x}_j) + (1 - y_j) L_0^\theta(\boldsymbol{x}_j) \right) + \frac{1}{|\mathcal{U}|} \sum_{i \in \mathcal{U}} L_0^\theta(\boldsymbol{x}_i). \quad (6)$$

As in ALOE both loss contributions are weighted equally. $L_{\text{Rand1}}(\theta)$ is minimized with respect to the backbone model parameters $\theta$.

- **Rand2.** The querying strategy of Trittenbach et al. (2021) samples uniformly among the top 50% data ranked by anomaly scores without replacement. This leades to a random set of "positive" queries. After the queries are labeled, the training loss function is the same as $L_{\text{Rand1}}(\theta)$ (Eq. (6)).

- **Mar.** After training the backbone model for one epoch, this querying strategy by Görnitz et al. (2013) uses the $\alpha$-quantile ($s_\alpha$) of the training data anomaly scores to define a "normality region". Then the $|\mathcal{Q}|$ samples closest to the margin $s_\alpha$ are selected to be queried. After the queries are labeled, the training loss function is the same as $L_{\text{Rand1}}(\theta)$ (Eq. (6)). Note that in practice we don't know the true anomaly ratio for the $\alpha$-quantile. In all experiment, we provide this querying strategy with the true contamination ratio of the dataset. Even with the true ratio, the "Mar" strategy is still outperformed by ALOE.

- **Hybr1.** This hybrid strategy, also used by (Görnitz et al., 2013) combines the "Mar" query with neighborhood-based diversification. The neighborhood-based strategy selects samples with fewer neighbors covered by the queried set to ensure the samples' diversity in the feature space. We start by selecting the data index $\arg\min_{1 \leq i \leq N} \|s_i - s_\alpha\|$ into $\mathcal{Q}$. Then the samples are selected sequentially without replacement by the criterion

$$\arg\min_{1 \leq i \leq N} 0.5 + \frac{|\{j \in \text{NN}_k(\phi(\boldsymbol{x}_i)) : j \in \mathcal{Q}\}|}{2k} + \beta \frac{\|s_i - s_\alpha\| - \min_i \|s_i - s_\alpha\|}{\max_i \|s_i - s_\alpha\| - \min_i \|s_i - s_\alpha\|}$$

where the inter-sample distance is measured in the feature space and the number of nearest neighbors is $k = \lceil N/|\mathcal{Q}| \rceil$. We set $\beta = 1$ for equal contribution of both terms. After the queries are labeled, the training loss function is the same as $L_{\text{Rand1}}(\theta)$ (Eq. (6)).

- **Pos1.** This querying strategy by Pimentel et al. (2020) always selects the top-ranked samples ordered by their anomaly scores, $\arg\max_{1 \leq i \leq N} s_i$. After the queries are labeled, the training loss only involves the labeled data

$$L_{\text{Pos1}}(\theta) = \frac{1}{|\mathcal{Q}|} \sum_{j \in \mathcal{Q}} \left( y_j L_1^\theta(\boldsymbol{x}_j) + (1 - y_j) L_0^\theta(\boldsymbol{x}_j) \right).$$

Pimentel et al. (2020) use the logistic loss but we use the supervised outlier exposure loss. The supervised outlier exposure loss is shown to be better than the logistic loss in learning anomaly detection models (Ruff et al., 2019; Hendrycks et al., 2018).

- **Pos2.** This approach of (Barnabé-Lortie et al., 2015) uses the same querying strategy as Pos1, but the training is different. Pos2 also uses the unlabeled data during training. After the queries are labeled, the training loss function is the same as $L_{\text{Rand1}}(\theta)$ (Eq. (6)).

- **Hybr2.** This hybrid strategy by Das et al. (2019) makes positive diverse queries. It combines querying according to anomaly scores with distance-based diversification. Hybr2 selects the initial query $\arg\max_{1 \leq i \leq N} s_i$ into $\mathcal{Q}$. Then the samples are selected sequentially without replacement by the criterion

$$\arg\max_{1 \leq i \leq N} \frac{s_i - \min_i s_i}{\max_i s_i - \min_i s_i} + \beta \min_{j \in \mathcal{Q}} \frac{d(\boldsymbol{x}_i, \boldsymbol{x}_j) - \min_{a \neq b} d(\boldsymbol{x}_a, \boldsymbol{x}_b)}{\max_{a \neq b} d(\boldsymbol{x}_a, \boldsymbol{x}_b) - \min_{a \neq b} d(\boldsymbol{x}_a, \boldsymbol{x}_b)}$$

where $d(\boldsymbol{x}_i, \boldsymbol{x}_j) = ||\phi(\boldsymbol{x}_i) - \phi(\boldsymbol{x}_j)||_2$. We set $\beta = 1$ for equal contribution of both terms. After the queries are labeled, Das et al. (2019) use the labeled set to learn a set of weights for the components of an ensemble of detectors. For a fair comparison of active learning strategies, we use the labeled set to update an individual anomaly detector with parameters $\theta$ by optimizing the loss

$$L_{\text{Hybr2}}(\theta) = \frac{1}{|\mathcal{Q}|} \sum_{j \in \mathcal{Q}} \left( y_j L_1^\theta(\boldsymbol{x}_j) + (1 - y_j) L_0^\theta(\boldsymbol{x}_j) \right).$$

- **Hybr3.** This baseline by (Ning et al., 2022) uses the same query strategy as Hybr2, but differs in the training loss function,

$$L_{\text{Hybr3}}(\theta) = \frac{1}{|\mathcal{Q}| + |\mathcal{U}|} \sum_{j \in \mathcal{Q}} w_j (1 - y_j) L_0^\theta(\boldsymbol{x}_j) + \frac{1}{|\mathcal{Q}| + |\mathcal{U}|} \sum_{i \in \mathcal{U}} \hat{w}_i L_0^\theta(\boldsymbol{x}_i),$$

where $w_j = 2\sigma(d_j)$ and $\hat{w}_i = 2 - 2\sigma(d_i)$ where $\sigma(\cdot)$ is the Sigmoid function and $d_i = 10c_d \left( ||\phi(\boldsymbol{x}_i) - \boldsymbol{c}_0||_2 - ||\phi(\boldsymbol{x}_i) - \boldsymbol{c}_1||_2 \right)$ where $\boldsymbol{c}_0$ is the center of the queried normal samples and $\boldsymbol{c}_1$ is the center of the queried abnormal samples in the feature space, and $c_d$ is the min-max normalization factor.

We make three observations for the loss function. First, $L_{\text{Hybr3}}(\theta)$ filters out all labeled anomalies in the supervised learning part and puts a large weight (but only as large as 2 at most) to the true normal data that has a high anomaly score. Second, in the unlabeled data, $L_{\text{Hybr3}}(\theta)$ puts smaller weight (less than 1) to the seemingly abnormal data. Third, overall, the weight of the labeled data is similar to the weight of the unlabeled data. This is unlike ALOE, which weighs labeled data $|\mathcal{U}|/|\mathcal{Q}|$ times higher than unlabeled data.

# D  IMPLEMENTATION DETAILS

In this section, we present the implementation details in the experiments. They include an overall description of the experimental procedure for all datasets, model architecture, data split, and details about the optimization algorithm.

## D.1  EXPERIMENTAL PROCEDURE

We apply the same experimental procedure for each dataset and each compared method. The experiment starts with an unlabeled, contaminated training dataset with index set $\mathcal{U}$. We first train the

anomaly detector on $\mathcal{U}$ for one epoch as if all data were normal. Then we conduct the diverse active queries at once and estimate the contamination ratio $\alpha$ by the importance sampling estimator Eq. (4). Lastly, we optimize the post-query training losses until convergence. The obtained anomaly detectors are evaluated on a held-out test set.

## D.2   DATA SPLIT

**Image Data.**   For the image data including both natural (CIFAR-10 (Krizhevsky et al., 2009) and F-MNIST (Xiao et al., 2017)) and medical (MedMNIST (Yang et al., 2021)) images, we use the original training, validation (if any), and test split. When contaminating the training data of one class, we randomly sample images from other classes' training data and leave the validation and test set untouched. Specifically for DermaMNIST in MedMNIST, we only consider the classes that have more than 500 images in the training data as normal data candidates, which include benign keratosis-like lesions, melanoma, and melanocytic nevi. We view all other classes as abnormal data. Different experiment runs have different randomness.

**Tabular Data.**   Our study includes the four multi-dimensional tabular datasets from the ODDS repository[2] which have an outlier ratio of at least $30\%$. . To form the training and test set for tabular data, we first split the data into normal and abnormal categories. We randomly sub-sample half the normal data as the training data and treat the other half as the test data. To contaminate training data, we randomly sub-sample the abnormal data into the training set to reach the desired $10\%$ contamination ratio; the remaining abnormal data goes into the test set. Different experiment runs have different randomness.

**Video Data.**   We use UCSD Peds1[3], a benchmark dataset for video anomaly detection. UCSD Peds1 contains 70 surveillance video clips – 34 training clips and 36 testing clips. Each frame is labeled to be abnormal if it has non-pedestrian objects and labeled normal otherwise. Making the same assumption as (Pang et al., 2020), we treat each frame independent and mix the original training and testing clips together. This results in a dataset of 9955 normal frames and 4045 abnormal frames. We then randomly sub-sample 6800 frames out of the normal frames and 2914 frames out of the abnormal frames without replacement to form a contaminated training dataset with $30\%$ anomaly ratio. A same ratio is also used in the literature (Pang et al., 2020) that uses this dataset. The remaining data after sampling is used for the testing set, whose about $30\%$ data is anomalous. Like the other data types, different experiment runs have different randomness for the training dataset construction.

## D.3   MODEL ARCHITECTURE

The experiments involve two anomaly detectors, NTL and multi-head RotNet (MHRot), and three data types.

**NTL on Image Data and Video Data.**   For all images (either natural or medical) and video frames, we extract their features by feeding them into a ResNet152 pre-trained on ImageNet and taking the penultimate layer output for our usage. The features are kept fixed during training. We then train an NTL on those features. We apply the same number of transformations, network components, and anomaly loss function $L_1^\theta(\boldsymbol{x})$, as when Qiu et al. (2022a) apply NTL on the image data.

**NTL on Tabular Data.**   We directly use the tabular data as the input of NTL. We apply the same number of transformations, network components, and anomaly loss function $L_1^\theta(\boldsymbol{x})$, as when Qiu et al. (2022a) apply NTL on the tabular data.

**MHRot on Image Data.**   We use the raw images as input for MHRot. We set the same transformations, MHRot architecture, and anomaly loss function as when Qiu et al. (2022a) apply MHRot on the image data.

---

[2]`http://odds.cs.stonybrook.edu/`
[3]`http://www.svcl.ucsd.edu/projects/anomaly/dataset.htm`

**DSVDD on Image Data.** For all images (either natural or medical), we build DSVDD on the features from the penultimate layer of a ResNet152 pre-trained on ImageNet. The features are kept fixed during training. The neural network of DSVDD is a three-layer MLP with intermediate batch normalization layers and ReLU activation. The hidden sizes are $[1024, 512, 128]$.

## D.4 OPTIMIZATION ALGORITHM

| Model | Dataset | Learning Rate | Epoch | Minibatch Size | $\tau$ |
|---|---|---|---|---|---|
| NTL | CIFAR-10 | 1e-4 | 30 | 512 | 1e-2 |
| | F-MNIST | 1e-4 | 30 | 512 | 1e-2 |
| | MedMNIST | 1e-4 | 30 | 512 | 1e-2 |
| | ODDS | 1e-3 | 100 | $\lceil N/5 \rceil$ | 1e-2 |
| | UCSD Peds1 | 1e-4 | 3* | 192 | 1e-2 |
| MHRot | CIFAR-10 | 1e-3 | 15 | 10 | N/A |
| | F-MNIST | 1e-4 | 15** | 10 | N/A |
| | MedMNIST | 1e-4 | 15 | 10 | N/A |
| Deep SVDD | CIFAR-10 | 1e-4 | 30 | 512 | 1e-2 |
| | F-MNIST | 1e-4 | 30 | 512 | 1e-2 |
| | MedMNIST | 1e-4 | 30 | 512 | 1e-2 |

*Hybr2, Hybr3, Pos1, and Pos2 train 30 epochs. All other methods train 3 epochs.
**ALOE train 3 epochs.

Table 5: A summary of optimization parameters for all methods.

In the experiments, we use Adam (Kingma and Ba, 2014) to optimize the objective function to find the local optimal anomaly scorer parameters $\theta$. For Adam, we set $\beta_1 = 0.9, \beta_2 = 0.999$ and no weight decay for all experiments.

To set the learning rate, training epochs, minibatch size for MedMNIST, we find the best performing hyperparameters by evaluating the method on the validation dataset. We use the same hyperparameters on other image data. For video data and tabular data, the optimization hyperparameters are set as recommended by Qiu et al. (2022a). We summarize all optimization hyperparameters in Tab. 5.

When training models with ALOE, we resort to the block coordinate descent scheme that update the model parameters $\theta$ and the pseudo labels $\tilde{y}$ of unlabeled data in turn. In particular, we take the following two update steps iteratively:

- update $\theta$ by optimizing Eq. (3) given $\tilde{y}$ fixed;

- update $\tilde{y}$ by sovling the constrained optimization in Sec. 3.3 given $\theta$ fixed;

Upon updating $\tilde{y}$, we use the $\text{LOE}_S$ variant (Qiu et al., 2022a) for the unlabeled data. We set the pseudo labels $\tilde{y}$ by performing the optimization below

$$\min_{\tilde{\boldsymbol{y}} \in \{0, 0.5\}^{|\mathcal{U}|}} \frac{1}{|\mathcal{U}|} \sum_{i \in \mathcal{U}} \tilde{y}_i L_1^\theta(\boldsymbol{x}_i) + (1 - \tilde{y}_i) L_0^\theta(\boldsymbol{x}_i) \qquad \text{s.t.} \qquad \sum_{i=1}^{|\mathcal{U}|} \tilde{y}_i = \frac{\tilde{\alpha}|\mathcal{U}|}{2},$$

where $\tilde{\alpha}$ is the updated contamination ratio of $\mathcal{U}$ after the querying round, $\tilde{\alpha} = \left( \alpha N - \sum_{j \in \mathcal{Q}} \mathbb{1}_a(\boldsymbol{x}_j) \right)/|\mathcal{U}|$, and $\alpha$ is computed by Eq. (4) given $\mathcal{Q}$. The solution is to rank the data by $L_0^\theta(\boldsymbol{x}) - L_1^\theta(\boldsymbol{x})$ and label the top $\tilde{\alpha}$ data abnormal (equivalently setting $\tilde{y} = 0.5$) and all the other data normal (equivalently $\tilde{y} = 0$).

When we compute the Euclidean distance in the feature space, we construct the feature vector of a sample by concatenating all its encoder representations of different transformations. For example, if the encoder representation has 500 dimensions and the model has 10 transformations, then the final feature representation has $10 \times 500 = 5000$ dimensions.

### D.5 TIME COMPLEXITY

Regarding the time complexity, the optimization uses stochastic gradient descent. The complexity of our querying strategy is $O(KN)$ where $K$ is the number of queries and $N$ is the size of the training data. This complexity can be further reduced to $O(K \log N)$ with a scalable extension of k-means++ (Bahmani et al., 2012).

## E  ADDITIONAL EXPERIMENTS AND ABLATION STUDY

The goal of this ablation study is to show the generality of ALOE, to better understand the success of ALOE, and to disentangle the benefits of the training objective and the querying strategy. To this end, we applied ALOE to different backbone models and different data forms (raw input and embedding input), performed specialized experiments to compare the querying strategies, to demonstrate the optimality of the proposed weighting scheme in Eq. (3), and to validate the detection performance of the estimated ratio by Eq. (4). We also compared ALOE against additional baselines including semi-supervised learning frameworks and shallow anomaly detectors.

### E.1  RESULTS WITH OTHER BACKBONE MODELS

Table 6: $|\mathcal{Q}| = 20$. AUC (%) with standard deviation for anomaly detection on six datasets (CIFAR-10, F-MNIST, Blood, OrganA, OrganC, OrganS). The backbone models are MHRot (Hendrycks et al., 2019) and Deep SVDD (Ruff et al., 2018). For all experiments, we set the contamination ratio as 10%. ALOE consistently outperforms two best-performing baselines on all six datasets.

|  | MHRot | | | Deep SVDD | | |
|---|---|---|---|---|---|---|
|  | ALOE | Hybr1 | Hybr2 | ALOE | Hybr1 | Hybr2 |
| CIFAR-10 | **86.9±0.7** | 83.9±0.1 | 49.1±2.0 | **93.1±0.2** | 89.0±0.6 | 91.3±1.0 |
| F-MNIST | **92.6±0.1** | 87.1±0.2 | 58.9±5.7 | **91.4±0.5** | 90.9±0.4 | 82.5±2.9 |
| Blood | **83.3±0.2** | 81.1±2.5 | 61.8±2.1 | **80.2±1.1** | 79.7±1.2 | 77.2±3.0 |
| OrganA | **96.5±0.3** | 94.1±0.3 | 61.1±4.8 | **89.5±0.3** | 87.1±0.7 | 71.3±3.8 |
| OrganC | **92.1±0.2** | 91.6±0.1 | 70.9±0.8 | **87.5±0.7** | 85.3±0.8 | 84.2±0.9 |
| OrganS | **89.3±0.2** | 88.3±0.3 | 68.2±0.1 | **85.5±0.7** | 83.4±0.3 | 81.2±1.3 |

We are interested whether ALOE works for different backbone models. To that end, we repeat part of the experiments in Tab. 2 but using an self-supervised learning model MHRot (Hendrycks et al., 2019) and a one class classification model Deep SVDD (Ruff et al., 2018) as the backbone model. We compare ALOE to two best performing baselines — Hybr1 and Hybr2. In this experiment, MHRot and Deep SVDD take different input types: while MHRot takes raw images as input, Deep SVDD uses pre-trained image features. We also set the query budget to be $|\mathcal{Q}| = 20$.

We report the results in Tab. 6. It showcases the superiority of ALOE compared to the baselines. On all datasets, ALOE significantly outperforms the two best performing baselines, Hybr1 and Hybr2, thus demonstrating the wide applicability of ALOE across anomaly detection model types.

### E.2  DISENTANGLEMENT OF ALOE

Table 7: $|\mathcal{Q}| = 20$. AUC (%) with standard deviation for anomaly detection on CIFAR-10 and F-MNIST. For all experiments, we set the contamination ratio as 10%. ALOE mitigates the performance drop when NTL and MHRot trained on the contaminated datasets. Results of the unsupervised method LOE are borrowed from Qiu et al. (2022a).

|  | NTL | | | MHRot | | |
|---|---|---|---|---|---|---|
|  | LOE | k-means++ | ALOE | LOE | k-means++ | ALOE |
| CIFAR-10 | 94.9±0.1 | 95.6±0.3 | **96.3±0.3** | 86.3±0.2 | 64.0±0.2 | **86.9±0.7** |
| F-MNIST | 92.5±0.1 | 94.3±0.2 | **94.8±0.4** | 91.2±0.4 | 91.5±0.1 | **92.6±0.1** |

We disentangle the benefits of each component of ALOE and compare it to unsupervised anomaly detection with latent outlier exposure (LOE) (Qiu et al., 2022a), and to supervised active anomaly detection with k-means++ querying strategy. Both active approaches (k-means++ and ALOE) are evaluated with $|\mathcal{Q}| = 20$ labeled samples. The unsupervised approach LOE requires an hyperparameter of the assumed data contamination ratio, which we set to the ground truth value $10\%$. Comparing ALOE to LOE reveals the benefits of the k-means++ active approach[4]; comparing ALOE to k-means++ reveals the benefits of the unsupervised loss function in ALOE. Results in Tab. 7 show that ALOE leads to improvements for both ablation models.

### E.3  COMPARISONS OF QUERYING STRATEGIES

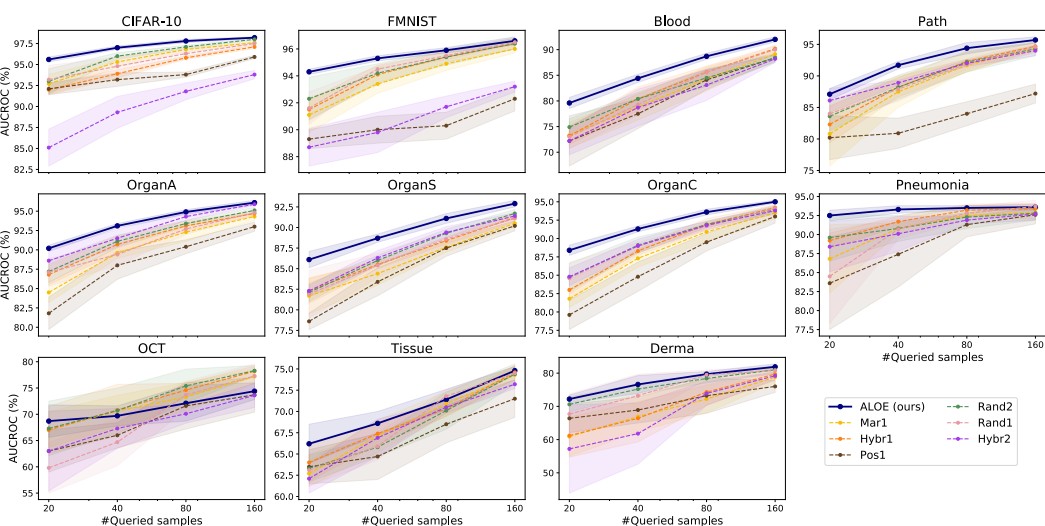

Figure 8: Ablation study on the query strategy. K-Means++ significantly outperforms other strategies for active anomaly detection on most of the datasets.

To understand the benefit of sampling diverse queries with k-means++ and to examine the generalization ability (stated in Thm. 1) of different querying strategies, we run the following experiment: We use a supervised loss on labeled samples to train various anomaly detectors. The only difference between them is the querying strategy used to select the samples. We evaluate them on all image data sets we study for varying number of queries $|\mathcal{Q}|$ between 20 and 160.

Results are in Fig. 8. On all datasets except OCT, k-means++ consistently outperforms all other querying strategies from previous work on active anomaly detection. The difference is particularly large when only few samples are queried. This also confirms that diverse querying generalizes better on the test data than other querying strategies (see additional results in Supp. A).

### E.4  ABLATION ON ESTIMATED CONTAMINATION RATIO

To see how the estimated ratio affects the detection performance, we compare ALOE to the counterpart with the true anomaly ratio. We experiment on all 11 image datasets. In Fig. 9, we report the average results for all datasets when querying $|\mathcal{Q}| = 20, 40, 80, 160$ samples. It shows that ALOE with either true ratio or estimated ratio performs similar given all query budgets. Therefore, the estimated ratio can be applied safely. This is very important in practice, since in many applications the true anomaly ratio is not known.

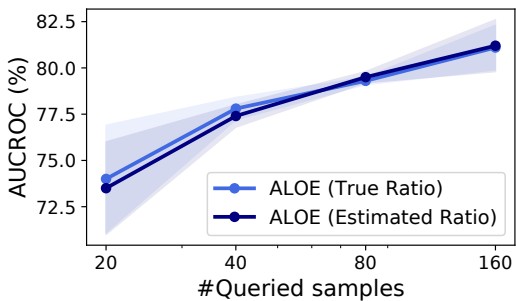

Figure 9: Model using the estimated ratio is indistinguishable from the one using the true ratio.

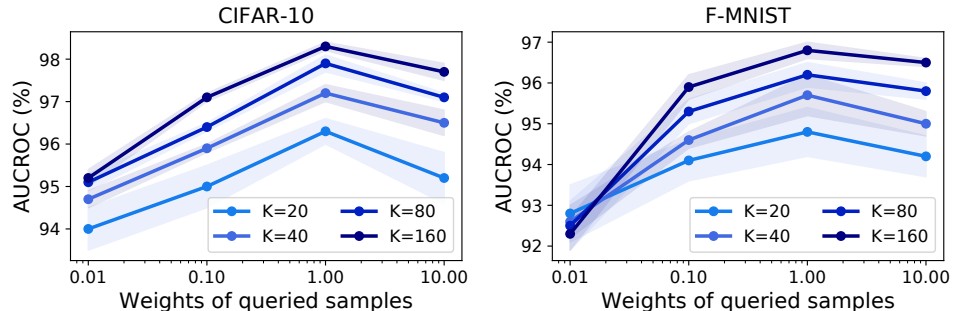

Figure 10: Ablation study on the weighting scheme in Eq. (3). With different query budgets $|\mathcal{Q}|$, the performance on image datasets degrades both upon down-weighting (0.01, 0.1) or up-weighting (10.0) the queried samples. In contrast, equal weighting yields optimal results.

### E.5 ABLATIONS ON WEIGHTING SCHEME

We make the implicit assumption that the *averaged* losses over queried and unqueried data should be equally weighted (Eq. (3)). That means, if a fraction $\epsilon$ of the data is queried, every queried data point weights $1/\epsilon$ as much as an unqueried datum. As a consequence, neither the queried nor the unqueried data points can dominate the result.

To test whether this heuristic is indeed optimal, we added a scalar prefactor to the supervised loss in Eq. (3) (the first term) and reported the results on the CIFAR-10 and F-MNIST datasets with different query budgets (Fig. 10). A weight<1 corresponds to downweighting the queried term, while a weight>1 corresponds to upweighting it. We use the same experimental setup and backbone (NTL) as in the paper. The results are shown in Fig. 10. We see that the performance degrades both upon down-weighting (0.01, 0.1) or up-weighting (10.0) the queried samples. In contrast, equal weighting yields optimal results.

### E.6 COMPARISONS WITH SEMI-SUPERVISED LEARNING FRAMEWORKS

ALOE exploits the unlabeled data to improve the model performance. This shares the same spirit of semi-supervised learning. We are curious about how a semi-supervised learning method performs in our active anomaly detection setup. To this end, we adapted an existing semi-supervised learning framework FixMatch (Sohn et al., 2020a) to our setup and compared with our method in Fig. 11. As follows, we will first describe the experiment results and then state the adaptation of FixMatch to anomaly detection we made.

FixMatch, as a semi-supervised learning algorithm, regularizes the image classifier on a large amount of unlabeled data. The regularization, usually referred to consistency regularization, requires the classifier to have consistent predictions on different views of unlabeled data, thus improves the

---

[4]Notice that while LOE uses the true contamination ratio (an oracle information), ALOE only uses the estimated contamination ratio by the 20 queries.

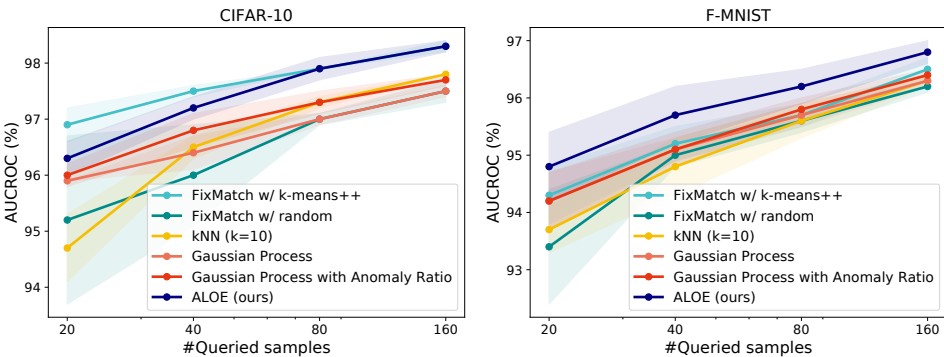

Figure 11: Comparison with semi-supervised learning fraemworks, FixMatch (Sohn et al., 2020a), $k$-nearest neighbors (Iscen et al., 2019), and Gaussian process (Li et al., 2018). On F-MNIST, ALOE outperforms all baselines, while on CIFAR-10, ALOE has a comparable performance with FixMatch with k-means++ querying.

classifier's performance. FixMatch generates various data views through image augmentations followed by Cutout (DeVries and Taylor, 2017). We noticed that, although FixMatch focuses on making use of the unlabeled data, its performance is highly affected by the quality of the labeled data subset. We investigated two variants depending on how we acquire the labeled data. One is the original semi-supervised learning setting, i.e., assuming the labeled data is a random subset of the whole dataset. The other one utilizes the same diversified data querying strategy k-means++ as ALOE to acquire the labeled part. In Fig. 11, we compared the performance of the two variants with ALOE. It shows that, on natural images CIFAR10 for which FixMatch is developped, while the original FixMatch with random labeled data is still outperformed by ALOE, FixMatch with our proposed querying strategy k-means++ has a comparable performance with ALOE. However, such advantage of FixMatch diminishes for the gray image dataset F-MNIST, where both variants are beat by ALOE on all querying budgets. In addition, the FixMatch framework is restrictive and may not be applicable for tabular data and medical data, as the augmentations are specially designed for natural images.

FixMatch is designed for classification. To make it suit for anomaly detection, we adapted the original algorithm[5] and adopted the following procedure and loss function.

1. Label all training data as normal and train the anomaly detector for one epoch;
2. Actively query a subset of data with size $|\mathcal{Q}|$, resulting in $\mathcal{Q}$ and the remaining data $\mathcal{U}$;
3. Finetune the detector in a supervised manner on non-augmented $\mathcal{Q}$ for 5 epochs;
4. Train the detector with the FixMatch loss Eq. (7) on augmented $\{\mathcal{U}, \mathcal{Q}\}$ until convergence.

We denote weak augmentation of input $\boldsymbol{x}$ by $\alpha(\boldsymbol{x})$ and the strong augmentation by $\mathcal{A}(\boldsymbol{x})$. The training objective function we used is

$$\mathcal{L}_{\text{FixMatch}}(\theta) = \frac{1}{|\mathcal{Q}|} \sum_{j \in \mathcal{Q}} \left( y_j L_1^\theta(\alpha(\boldsymbol{x}_j)) + (1 - y_j) L_0^\theta(\alpha(\boldsymbol{x}_j)) \right)$$
$$+ \frac{1}{|\mathcal{U}|} \sum_{i \in \mathcal{U}} \mathbb{1}(S(\alpha(\boldsymbol{x}_i)) < q_{0.7} \text{ or } S(\alpha(\boldsymbol{x}_i)) > q_{0.05}) \left( \tilde{y}_i L_1^\theta(\mathcal{A}(\boldsymbol{x}_i)) + (1 - \tilde{y}_i) L_0^\theta(\mathcal{A}(\boldsymbol{x}_i)) \right) \quad (7)$$

where pseudo labels $\tilde{y}_i = \mathbb{1}(S(\alpha(\boldsymbol{x}_i)) > q_{0.05})$ and $q_n$ is the $n$-quantile of the anomaly scores $\{S(\alpha(\boldsymbol{x}_i))\}_{i \in \mathcal{U}}$. In the loss function, we only use the unlabeled samples with confidently predicted pseudo labels. This is controlled by the indicator function $\mathbb{1}(S(\alpha(\boldsymbol{x}_i)) < q_{0.7} \text{ or } S(\alpha(\boldsymbol{x}_i)) > q_{0.05})$. We apply this loss function for mini-batches on a stochastic optimization basis.

We also extend the semi-supervised learning methods using non-parametric algorithms to our active anomaly detection framework. We applied $k$-nearest neighbors and Gaussian process for inferring

---

[5]We adapted the FixMatch implementation `https://github.com/kekmodel/FixMatch-pytorch`

the latent anomaly labels (Iscen et al., 2019; Li et al., 2018) because these algorithms are unbiased in the sense that if the queried sample size is large enough, the inferred latent anomaly labels approach to the true anomaly labels. For these baselines, we also queried a few labeled data with k-means++ -based diverse querying strategy and then annotate the unqueried samples with k-nearest neighbor classifier or Gaussian process classifer trained on the queried data.

Both methods become ablations of ALOE. We compare ALOE with them on CIFAR-10 and F-MNIST under various query budgets and report their results in Fig. 11. On both datasets, ALOE improves over the variant of using only queried samples for training. On F-MNIST, ALOE outperforms all ablations clearly under all query budgets, while on CIFAR-10, ALOE outperforms all ablations except for FixMatch when query budget is low. In conclusion, ALOE boosts the performance by utilizing the unlabeled samples properly, while other labeling strategies are less effective.

### E.7 MORE COMPARISONS

Table 8: Comparisons with kNN method. We reported the F1-score (%) with standard error for anomaly detection on tabular datasets when the query budget $K = 10$. ALOE outperforms the kNN baseline.

|  | $k^{\text{th}}$NN | ALOE |
|---|---|---|
| **BreastW** | 92.5±2.1 | **93.9±0.5** |
| **Ionosphere** | 88.1±1.3 | **91.8±1.1** |
| **Pima** | 40.5±4.7 | **55.5±1.2** |
| **Satellite** | 61.1±2.2 | **71.1±1.7** |
| **Average** | 70.6 | **78.1** |

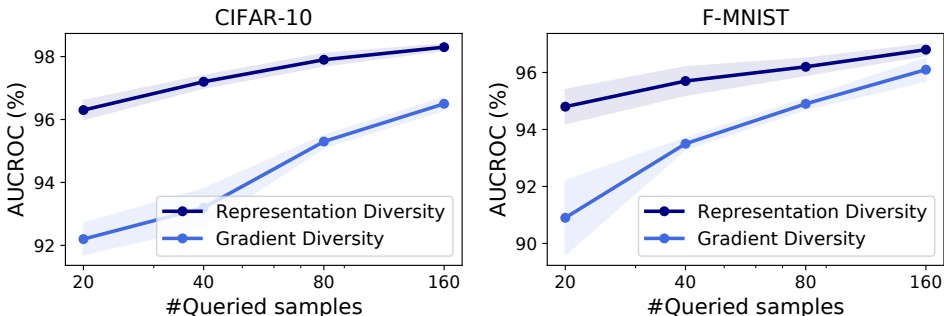

Figure 12: Comparison with gradient diversity querying strategy (BADGE) (Ash et al., 2020). The gradients wrt. the penultimate layer representation don't provide as informative queries as the representation itself, thus outperformed by our querying strategy ALOE. The true contamination ratio is 10%.

**Comparisons to kNN (Ramaswamy et al., 2000)** We compared against kNN in two ways. First we confirmed that our baseline backbone model NTL is competitive with kNN, which is shown to have a strong performance on tabular data (Shenkar and Wolf, 2022). To this end, NTL has been shown to yield 95.7% AUC on clean CIFAR-10 data, see Shenkar and Wolf, 2022, Table 1 column 1. In contrast, Qiu et al. (2022a) reported 96.2% AUC in Table 2, which is very close.

Second, we tested the performance of the kNN method on our corrupted training data set. We gave kNN the advantage of using the ground truth contamination ratio (otherwise when under-estimating this value, we saw the method degrade severely in performance).

KNN has two key hyperparameters: the number of nearest neighbors $k$ and the assumed contamination ratio of the training set. The method uses this assumed contamination ratio when fitting to define the threshold on the decision function. In our experiments, we tried multiple values of $k$ and reported the best testing results. Although the ground truth anomaly rate is unknown and our proposed methods don't have access to it, we gave kNN the competitive advantage of knowing the ground truth contamination ratio.

We studied the same tabular data sets as in our paper: BreastW, Ionosphere, Pima, and Satellite. We used the same procedure for constructing contaminated data outlined in our paper, where the contamination ratio was set to 10%. The results are summarized in Tab. 8.

We adopted PyOD's implementation of kNN[6] and set all the other hyperparameters to their default values ("method, radius, algorithm, leaf_size, metric, p, and metric_params"). We repeated the experiments 10 times and reported the mean and standard deviation of the F1 scores in Tab. 8. We find that our active learning framework outperforms the kNN baseline.

In more detail, the F1 scores for different values of $k$ are listed below, where $k = 1, 2, 5, 10, 15, 20$, respectively:

- BreastW: 84.3±7.6, 86.5±3.1, 89.9±3.9, 90.7±3.4, 92.5±2.1, 91.9±1.5
- Ionosphere: 88.1±1.3, 87.6±2.6, 84.5±3.9, 75.2±2.5, 70.4±3.6, 67.4±3.4
- Pima: 34.4±3.6, 32.3±3.4, 36.9±6.4, 40.5±4.7, 35.3±3.6, 35.5±4.5
- Satellite: 51.0±1.1, 53.5±0.7, 54.7±1.3, 57.4±1.8, 59.3±1.3, 61.1±2.2

**Comparisons to Gradient Diversity Querying Strategy (BADGE) (Ash et al., 2020)** We compared against a popular active learning method, BADGE (Ash et al., 2020), which is a diversity-driven active learning method that exploits sample-wise gradient diversity. We start with observing that BADGE doesn't work well for anomaly detection in Fig. 12, where we only replaced the objects that k-means++ works on in ALOE with gradients demanded in BADGE (Ash et al., 2020) while keeping all other settings fixed. This variant is referred to as "Gradient Diversity" while ours is denoted by "Representation Diversity". Fig. 12 shows the performance of Gradient Diversity is outperformed by a large margin, failing in querying informative samples as our Representation Diversity.

To understand which part of BADGE breaks for anomaly detection tasks, we check the gradients used by BADGE in an anomaly detection model. Before that, we start with describing how BADGE works. BADGE is developed for active learning in classification tasks. Given a pre-trained classifier, it first predicts the most likely label $\hat{y}$ (pseudo labels) for the unlabeled training data $\boldsymbol{x}$. These pseudo labels are then used to formulate a cross entropy loss $l_{CE}(\boldsymbol{x}, \hat{y})$. BADGE computes every data point's loss function's gradient to the final layer's weights as the data's representation. Upon active querying, a subset of data are selected such that their representations are diverse. In particular, the gradient to each class-specific weight $W_k$ is $\nabla_{W_k} l_{CE}(\boldsymbol{x}, \hat{y}) = (p_k - \mathbb{1}(\hat{y} = k))\phi(\boldsymbol{x})$ where $p_k$ is the predicted probability of being class $k$ and $\phi(\boldsymbol{x})$ is the output of the penultimate layer. Proposition 1 of Ash et al. (2020) shows the norm of the gradient with pseudo labels is a lower bound of the one with true labels. In addition, note that the gradient is a scaling of the penultimate layer output. The scaling factor describes the predictive uncertainty and is upper bounded by 1. Therefore, the gradients are informative surrogates of the penultimate layer output of the network, as shown by the inequality

$$||\nabla_{W_k} l_{CE}(\boldsymbol{x}, \hat{y})||^2 \leq ||\nabla_{W_k} l_{CE}(\boldsymbol{x}, y)||^2 \leq ||\phi(\boldsymbol{x})||^2. \tag{8}$$

However, these properties are associated with the softmax activation function usage. In anomaly detection, models and losses are diverse and are beyond the usage of softmax activation outputs. Hence the gradients are no longer good ways to construct active queries. For example, the supervised deep SVDD (Ruff et al., 2019) uses the contrasting loss $l(\boldsymbol{x}, y) = y/(W\phi(\boldsymbol{x}) - \boldsymbol{c})^2 + (1-y)(W\phi(\boldsymbol{x}) - \boldsymbol{c})^2$ to compact the normal sample representations around center $\boldsymbol{c}$. However, the gradient $\nabla_W l(\boldsymbol{x}, y) = \left(2(1-y)(W\phi(\boldsymbol{x}) - c) - 2y(W\phi(\boldsymbol{x}) - c)^{-3}\right)\phi(\boldsymbol{x})$ is not a bounded scaling of $\phi(\boldsymbol{x})$ any more, thus not an informative surrogate of point $\boldsymbol{x}$.

### E.8 NTL as a Unified Backbone Model

In Section 4 of the main paper, we have empirically compared ALOE to active-learning strategies known from various existing papers, where these strategies originally were proposed using different backbone architectures (either shallow methods or simple neural architectures, such as autoencoders). However, several recent benchmarks have revealed that these backbones are no longer competitive with modern self-supervised ones (Alvarez et al., 2022). For a fair empirical comparison of ALOE to modern baselines, we upgraded the previously proposed active-learning methods by replacing their

---

[6]https://github.com/yzhao062/pyod

simple respective backbones with a modern self-supervised backbone: NTL (Qiu et al., 2021)—the same backbone that is also used in ALOE.

We motivate our choice of NTL as unified backbone in our experiments as follows. Fig. 13 shows the results of ten shallow and deep anomaly detection methods (Qiu et al., 2022a; Deecke et al., 2018; Ruff et al., 2018; Golan and El-Yaniv, 2018; Hendrycks et al., 2019; Tax and Duin, 2004; Liu et al., 2008; Diederik P. Kingma, 2014; Makhzani and Frey, 2015; Sohn et al., 2020b) on the CIFAR10 one-vs.-rest anomaly detection task. NTL performs best (by a large margin) among the compared methods, including many classic backbone models known from the active anomaly detection literature (Trittenbach et al., 2021; Ruff et al., 2019; Görnitz et al., 2013; Das et al., 2019; Pimentel et al., 2020; Ning et al., 2022; Barnabé-Lortie et al., 2015).

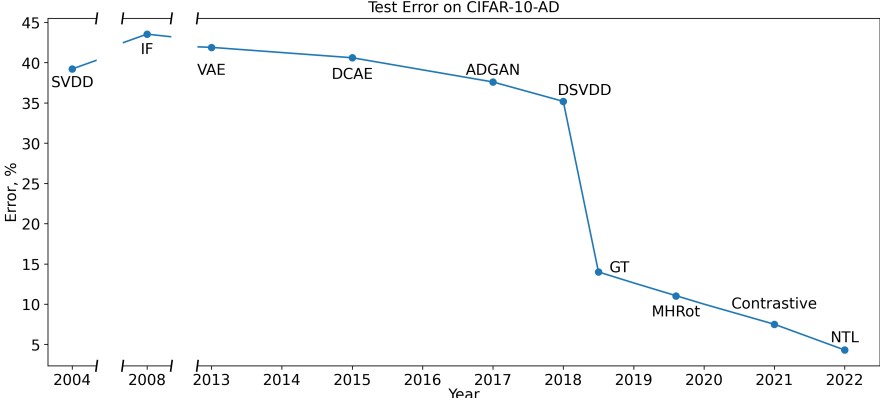

Figure 13: Error (in % of 1-AUCROC) of ten methods on CIFAR10: two shallow methods (SVDD (Tax and Duin, 2004) and IF (Liu et al., 2008)) and eight deep methods (VAE (Diederik P. Kingma, 2014), DCAE (Makhzani and Frey, 2015), ADGAN (Deecke et al., 2018), DSVDD (Ruff et al., 2018), GT (Golan and El-Yaniv, 2018), MHRot (Hendrycks et al., 2019), Contrastive (Sohn et al., 2020b), and NTL (Qiu et al., 2022a)). NTL achieves the best anomaly detection performance on CIFAR10.

Table 9: F1-scores (in %) and their standard deviations of 13 anomaly detection methods on tabular data. Results are taken from Alvarez et al. (2022). The results indicate that NTL is the state-of-the-art for tabular anomaly detection.

|          | KDDCUP10 | NSL-KDD | IDS2018 | Arrhythmia | Thyroid | Avg. |
|----------|----------|---------|---------|------------|---------|------|
| ALAD       | 95.9±0.7 | 92.1±1.5 | 59.0±0.0 | 57.4±0.4 | 68.6±0.5 | 74.6 |
| DAE        | 93.2±2.0 | **96.1±0.1** | **71.5±0.5** | 61.5±2.5 | 59.0±1.5 | 76.3 |
| DAGMM      | 95.9±1.4 | 85.3±7.4 | 55.8±5.3 | 50.6±4.7 | 48.6±8.0 | 67.2 |
| DeepSVDD   | 89.1±2.0 | 89.3±2.0 | 20.8±11 | 55.5±3.0 | 13.1±13 | 53.6 |
| DROCC      | 91.1±0.0 | 90.4±0.0 | 45.6±0.0 | 35.8±2.6 | 62.1±10 | 65.0 |
| DSEBM-e    | 96.6±0.1 | 94.6±0.1 | 43.9±0.8 | 59.9±1.0 | 23.8±0.7 | 63.8 |
| DSEBM-r    | **98.0±0.1** | 95.5±0.1 | 40.7±0.1 | 60.1±1.0 | 23.6±0.4 | 63.6 |
| DUAD       | 96.5±1.0 | 94.5±0.2 | **71.8±2.7** | 60.8±0.4 | 14.9±5.5 | 67.7 |
| MemAE      | 95.0±1.7 | 95.6±0.0 | 59.9±0.1 | 62.6±1.6 | 56.1±0.9 | 73.8 |
| SOM-DAGMM  | 97.7±0.3 | 95.6±0.3 | 44.1±1.1 | 51.9±5.9 | 52.7±12 | 68.4 |
| LOF        | 95.1±0.0 | 91.1±0.0 | 63.8±0.0 | 61.5±0.0 | 68.6±0.0 | 76.0 |
| OC-SVM     | 96.7±0.0 | 93.0±0.0 | 45.4±0.0 | **63.5±0.0** | 68.1±0.0 | 73.3 |
| NTL        | 96.4±0.2 | **96.0±0.1** | 59.5±8.9 | 60.7±3.7 | **73.4±0.6** | **77.2** |

An independent benchmark comparison of 13 methods (including nine deep methods proposed in 2018–2022) (Alvarez et al., 2022) recently identified NTL as the leading anomaly-detection method on tabular data. In their summary, the authors write: 'NeuTraLAD, the transformation-based approach, offers consistently above-average performance across all datasets. The data-augmentation strategy is particularly efficient on small-scale datasets where samples are scarce.'. Note that the

latter is also the scenario where active learning is thought to be the most promising. We show the results from Alvarez et al. (2022) in Tab. 9.

