# OpenReview forum: "Deep Active Anomaly Detection With Diverse Queries"
_ICLR.cc/2023/Conference — Submitted to ICLR 2023_

### Official Review · Reviewer_7TvR · 2022-10-23

**Confidence:** 4
**Correctness:** 3
**Technical Novelty And Significance:** 3
**Empirical Novelty And Significance:** 3
**Recommendation:** 6

**Clarity, Quality, Novelty And Reproducibility:**

This paper is well written and it is easy to follow. Meanwhile, a novel active learning strategy is proposed and termed as active latent outlier exposure (ALOE). The parameters in ALOS are adaptively designed.

**Strength And Weaknesses:**

Strength:
1.	The proposed method can fully exploits the supervised and unsupervised information in anomaly detection.
2.	The important hyperparameters in the proposed method are designed adaptively.
3.	The theorems and assumptions used in the paper are theoretically derived and experimentally verified in detail and the experimental results of the proposed method are promising.

Weakness:
1)	The time complexity of the proposed method is not clearly. In the experiment, why the proposed method is not verified on large-scale dataset.
2)	In the proposed method, a querying strategy that encourages diversity with k-means++ is proposed method. However, the main contributions of this strategy are not clearly presented.
3)	In the proposed method, the equation(3) mainly deals with binary classification. How does the proposed method deal with the multiple classification setting.



**Summary Of The Paper:**

This paper proposed an active learning approach for deep anomaly detection where a k-means++ based diversified querying strategy is adopted to ensure that the queries cover both the normal data and the anomalies well and two losses for queried and unqueried samples are designed to prevent neither the queried nor the unqueried data points from dominating the learning task. This paper also provides a method to estimate the unknown contamination rate from queried samples thus avoiding the assignment of the most important hyperparameters. This paper also provides a detailed theoretical derivation and experimental verification of the assumptions and theories used in the method. The experimental results show that the proposed method outperforms the compared methods significantly. The ablation study also shows the effectiveness of the proposed method.


**Summary Of The Review:**

This paper proposed an active learning approach for deep anomaly detection which is composed of unsupervised anomaly detection with latent outlier exposure (LOE) and supervised active anomaly detection with k-means++ querying strategy. It fully exploits the supervised and unsupervised information into anomaly detection, which ensures the diversity of the selected samples and the generalization of the scoring function to unlabeled and labeled sets. The theoretical derivations and experiments in the article are very detailed. Overall the article is novel enough and well-written.

---

> ### Author Response · Authors · 2022-11-10
> **Response to Reviewer 7TvR**
>
> We thank the reviewer for the valuable feedback. Our responses to the comments are as follows.
>
> > “The time complexity of the proposed method is not clearly. In the experiment, why the proposed method is not verified on large-scale dataset.”
>
> The datasets used in our paper are used as standard benchmarks in the community and they have been used virtually in all top-tier contributions in the field [Ruff et al., 2021]. We evaluated the proposed method on a variety of data types and datasets. Regarding the time complexity, the optimization uses stochastic gradient descent. The complexity of our querying strategy is O(KN) where K is the number of queries and N is the size of the training data. This complexity can be further reduced to O(KlogN) with a scalable extension of Kmeans++ [Bahmani et al., 2012].
>
> Ruff, Lukas, et al. "A unifying review of deep and shallow anomaly detection." Proceedings of the IEEE 109.5 (2021): 756-795.
>
> Bahmani, Bahman, et al. "Scalable K-Means++." Proceedings of the VLDB Endowment 5.7 (2012).
>
> > “2) In the proposed method, a querying strategy that encourages diversity with k-means++ is proposed method. However, the main contributions of this strategy are not clearly presented.”
>
> Our contribution lies in the theoretical analysis (see our general comments). Based on the analysis, we proposed a diversity-driven querying strategy, which uses Kmeans++ as a design choice. That said, using diverse queries is novel in active (deep) anomaly detection and our strategy outperforms existing works both theoretically and empirically.
>
> > “3) In the proposed method, the equation(3) mainly deals with binary classification. How does the proposed method deal with the multiple classification setting.”
>
> We believe this is a misunderstanding. We label the normal data as 0 and all the abnormal data as 1. We never use any class labels.

---

### Official Review · Reviewer_uxaT · 2022-10-27

**Confidence:** 3
**Correctness:** 2
**Technical Novelty And Significance:** 1
**Empirical Novelty And Significance:** 2
**Recommendation:** 3

**Clarity, Quality, Novelty And Reproducibility:**

The paper is well-written, but lacks novelty. Classic technique from classification based active learning has been adopted.

**Strength And Weaknesses:**

1. Abstract: "However, correctly identifying anomalies requires an estimate of the fraction of anomalies in the data." -- This has never been proven (to my knowledge). While specific algorithms might make assumptions, this is not true in general. Anomalies are by nature unknown unknowns. Even their fraction is not knowable. The paper probably makes this statement w.r.t the specific LOE algorithm discussed and should clarify that or remove this statement.


2. Section 2: "... not comparable to Pelleg and Moore (2004), Siddiqui et al. (2018), and Ghasemi et al. (2011), who fit a density model on the raw input data ..." -- Siddiqui et al. is tree-based. They do not fit a density model.


3. Section 3.2: "... determined by the smallest radius \delta such that ..." -- Is the radius \delta here an absolute distance measure, or in normalized units? It does not make sense if the distance is unnormalized and if it (distance between data points) could vary from zero to infinity.


4. Section 3.2: "... all unlabeled points are within distance \delta of one queried sample of the same type." -- This makes strong and oversimplifying assumption about the data. It basically means assuming that data from various classes are tightly clustered together.


5. Section 3.2: "... we propose a querying strategy that encourages diversity: k-means++ ..." -- Querying cluster centers is a classic active query strategy and not novel. For example, [r1] proposes an approach to query the centroids of clusters of instances that lie closest to the decision boundary.


6. Assumption 2: I suspect Assumption 2 has fallen for a statistical fallacy.

As an analogy to the Assumption 2 made here, suppose we consider undergrad students at a top ranked US university. The average (say) SAT scores will be very high and the distribution of the scores around that (high) average will be statistical noise -- just like the off-diagonal correlation mentioned here. However, that does not mean that the distribution of SAT scores among the general population will be the same as at the university.


7. Section 4: "... and all |Q| queries are collected at once." -- This is not really an active learning setup. For comparing with active learning algorithms on a fair basis, the algorithm should iterate over multiple label feedbacks from a (simulated) human-in-the-loop.


8. Table 2: "For all experiments, we set the contamination ratio as 10%." -- Experiments should be run with actual contamination factor in the datasets, not the artificial 10%.


9. Section 4.2: "... which have an outlier ratio of at least 30%." -- This amount of anomalies does not really make it an anomaly detection problem. The set of experiments are missing some harder real-world datasets where the anomaly fraction is really small ~1-2%.


10. Appendix C: "Hybr2. This hybrid strategy by Das et al. (2019) makes positive diverse queries." -- The main query strategy for diversity in Das et al. (2019) is based on compact description. The strategy explained here is likely one of the alternatives proposed on the related github repo (https://github.com/shubhomoydas/ad_examples). This needs to be clarified. It should be noted that the compact description based approach in Das et al. is basically a clustering-based approach.


References:

[r1] Xu, Z., Akella, R., & Zhang, Y. (2007). Incorporating Diversity and Density in Active Learning for Relevance Feedback. ECIR.


>>>>>>>>>>> Update after author response

I thank the authors for responding to my comments. At this time I do not have sufficient reason to change my scores. Mainly, I do not see this as a true 'active learning' anomaly detection algorithm and it comes across as mostly a class-imbalanced classification problem. The paper started off with an unproven and unjustified premise that correctly identifying anomalies requires estimating the anomaly fraction. It appears that the premise might have some bearing under a few strong assumptions for a specific anomaly detection algorithm, but that is not generalizable. Some more comments:


1. Clarification on Assumption 2 (and also Assumption 1) comment: The paper assumes that if off-diagonal elements of the score matrix constructed from a subset of data are zero, then you can deduce some properties of the general population from that. But, that is wrong as I pointed out with an example. You can have query schemes where the off diagonal correlations can be zero, but the general population's score distribution cannot be deduced from that. That is why the theory is not well-grounded.


2. "Different application areas require a different active learning setup.... The experimental comparison is fair because all methods are evaluated in the same setting..." -- This is where we have a contradiction. Most (all?) of the competitor/benchmark active learning algorithms used in this paper were designed for continuous active learning. So then why compare a one-shot feedback algorithm with the continuous feedback algorithms?


3. "This video dataset is a benchmark dataset for anomaly detection and has the original outlier ratio of around 30%." -- In that case let's use more / other datasets where the anomaly ratios are very small.


4. "We refer to their strategy based on the description in the paragraph 'Diversity-based Query Selection Strategy' on page 12" -- The method described in the current work is not the same as Das et al. (2019) and should be made clear.


5. "..reviewer i6D3 found our theoretical study novel and reviewer 7TvR said.." -- My review not being in agreement with other reviewers is (among other things) a sign of having an independent opinion -- which I last checked is highly valued among reviewers at ICLR.


**Summary Of The Paper:**

The paper proposes a diversified query strategy for an anomaly detector that can be applied in an active learning setup. The querying strategy is based on querying diverse cluster centers seeded by k-means++.

**Summary Of The Review:**

The theory and assumptions are not well grounded. The claim that this is in an active learning setting does not match the experiment setup where instead of multiple rounds of feedback, all query instances are selected at one shot.

---

> ### Author Response · Authors · 2022-11-10
> **Response to Reviewer uxaT (1)**
>
> We thank the reviewer for the detailed comments and valuable suggestions. We address them below.
>
> > “1. Abstract: However, correctly identifying anomalies requires an estimate of the fraction of anomalies in the data...The paper probably makes this statement w.r.t the specific LOE algorithm discussed and should clarify that or remove this statement.”
>
> We agree that we should clarify that many algorithms, when working with contaminated data, rely on a hyperparameter called the contamination ratio which corresponds to the fraction of the anomalies in the data. For example, OC-SVM [Schölkopf et al., 1999], kNN [Ramaswamy et al., 2000], Robust PCA, Robust Deep Auto-encoder [Zhou et al., 2017], and soft-boundary deep SVDD [Ruff et al., 2018] require a decent estimate of the contamination ratio for good performance. These methods also use the contamination ratio to identify the threshold on the anomaly scores (only samples with scores above the threshold are considered anomalies). Thus estimating the contamination ratio is an important problem with potential benefits for many anomaly detection algorithms. We will clarify that statement in our paper.
>
> Schölkopf, Bernhard, et al. "Support vector method for novelty detection." Advances in neural information processing systems 12 (1999).
>
> Ramaswamy, S., Rastogi, R. and Shim, K., 2000, May. Efficient algorithms for mining outliers from large data sets. ACM Sigmod Record, 29(2), pp. 427-438.
>
> Zhou, Chong, and Randy C. Paffenroth. "Anomaly detection with robust deep autoencoders." Proceedings of the 23rd ACM SIGKDD international conference on knowledge discovery and data mining. 2017.
>
> > “2. Section 2: "... not comparable to Pelleg and Moore (2004), Siddiqui et al. (2018), and Ghasemi et al. (2011), who fit a density model on the raw input data ..." -- Siddiqui et al. is tree-based. They do not fit a density model.”
>
> Thank you for correcting us! We will remove “Siddiqui et al. (2018)” from this sentence. We also mentioned Siddiqui et al. (2018)’s most-positive querying strategy in Sec. 2.
>
> > “3. Section 3.2: "... determined by the smallest radius \delta such that ..." -- Is the radius \delta here an absolute distance measure, or in normalized units? It does not make sense if the distance is unnormalized and if it (distance between data points) could vary from zero to infinity.”
>
> The radius is measured by Euclidean distance in the feature space. This radius cannot go to infinity because we are restricted to *finite* training data in practice. Fig. 3 in Supp. A shows the definition of radius (Eq. 5) is feasible.
>
> > “4. Section 3.2: "... all unlabeled points are within distance \delta of one queried sample of the same type." -- This makes strong and oversimplifying assumption about the data. It basically means assuming that data from various classes are tightly clustered together. ”
>
> We respectfully disagree. The computation of the radius doesn’t require any such assumption and the definition (rigorously Eq. 5) applies in a generic way. By saying “type”, we mean “normal” or “abnormal” and we don’t use any data class information in the definition. In the experiments, we employ the one-vs-rest setup, which ensures the anomalies are diverse. Fig. 3 plots the computed radius for every querying strategies with varying query budgets, which shows Eq. 5 is a feasible definition.
>
> > “5. Section 3.2: "... we propose a querying strategy that encourages diversity: k-means++ ..." -- Querying cluster centers is a classic active query strategy and not novel. For example, [r1] proposes an approach to query the centroids of clusters of instances that lie closest to the decision boundary.”
>
> Thank you for mentioning the classic diversity-based active querying strategy. Our proposed querying strategy is a specific design choice. Our contribution lies in the proposed theory (Thm. 1) of analyzing different querying strategies tailored for anomaly detection. Based on the theory we illustrate why diverse queries generalize better than other types of queries. We also empirically demonstrate that diverse querying outperforms existing work using positive queries or uncertain queries, so our contribution is both theoretical and empirical.

---

> ### Author Response · Authors · 2022-11-10
> **Response to Reviewer uxaT (2)**
>
> > “6. Assumption 2: I suspect Assumption 2 has fallen for a statistical fallacy.”
>
> We don’t see the analogy to Assumption 2 and respectfully ask for further clarification. Assumption 2 says the anomaly scores are predictive to the distribution of both the queried data and the original data, which is empirically tested in Supp. B.2. This assumption is used to construct an importance-weighted estimator of the contamination ratio. We tested the robustness of the estimator in Supp. B.3, where we estimated multiple ground truth anomaly ratios (1%, 5%, 10%, 15%). The results show that most estimates are within the error bars and hence accurate. The estimation error is acceptable, as confirmed by the sensitivity study in (Qiu et al., 2022a) which concludes that the LOE approach still works well if the anomaly ratio is mis-specified within 5 percentage points.
>
> > “7. Section 4: "... and all |Q| queries are collected at once." -- This is not really an active learning setup. For comparing with active learning algorithms on a fair basis, the algorithm should iterate over multiple label feedbacks from a (simulated) human-in-the-loop.” “The claim that this is in an active learning setting does not match the experiment setup where instead of multiple rounds of feedback, all query instances are selected at one shot.”
>
> Different application areas require a different active learning setup. In some settings, active learning involves a human in-the-loop. In the batch active learning setting that we study in this paper, the expert is asked only once to label an entire batch. We will describe this setting more clearly in the paper. The experimental comparison is fair because all methods are evaluated in the same setting. Regarding the performance vs. the number of queries, Fig. 1 and Fig. 2 demonstrate their relation. It shows that the performance improves with the number of queries, and that consistent with our theory, diverse querying makes the best use of low query budgets.
>
> > “8. Table 2: "For all experiments, we set the contamination ratio as 10%." -- Experiments should be run with actual contamination factor in the datasets, not the artificial 10%.”
>
> Consistent with previous work, an anomaly detection benchmark is created synthetically from a classification dataset using the one-vs.-rest setup described in Sec. 4.1 . To create the contaminated training data, we mix the data of the class considered normal with anomalies from other classes and therefore have full control over the contamination ratio.
>
> > “9. Section 4.2: "... which have an outlier ratio of at least 30%." -- This amount of anomalies does not really make it an anomaly detection problem. The set of experiments are missing some harder real-world datasets where the anomaly fraction is really small ~1-2%.”
>
> This video dataset is a benchmark dataset for anomaly detection and has the original outlier ratio of around 30%. See Pang et al. (2020) and the data description (http://www.svcl.ucsd.edu/projects/anomaly/dataset.htm).
>
> > “10. Appendix C: "Hybr2. This hybrid strategy by Das et al. (2019) makes positive diverse queries." -- The main query strategy for diversity in Das et al. (2019) is based on compact description. The strategy explained here is likely one of the alternatives proposed on the related github repo (https://github.com/shubhomoydas/ad_examples). This needs to be clarified. It should be noted that the compact description based approach in Das et al. is basically a clustering-based approach.”
>
> We refer to their strategy based on the description in the paragraph “Diversity-based Query Selection Strategy” on page 12 of Das et al. (2019). Based on their description, Das et al. (2019) encouraged diversity on top of positive queries. Because we want to compare different strategies on a fair basis, we apply the NTL or Deep SVDD as a backbone anomaly detector instead of an isolation forest, to which the term “compact description” is related.
>
> > “The paper is well-written, but lacks novelty. Classic technique from classification based active learning has been adopted.”
>
> We did a novel theoretical analysis specifically for anomaly detection and, based on the theory, proposed a diverse querying strategy in this area for the first time. Our querying strategy is a design choice. We empirically demonstrated diverse querying outperforms existing works in this area.
>
> > “The theory and assumptions are not well grounded.”
>
> We respectfully disagree. We provided all derivations and empirically verified the assumptions in the paper. In addition, reviewer i6D3 found our theoretical study novel and reviewer 7TvR said “the theorems and assumptions used in the paper are theoretically derived and experimentally verified in detail.”

---

> > ### Author Response · Authors · 2022-11-18
> > **(Round 2) Batch active anomaly detection; comparison to class-imbalanced classification; anomaly ratio estimation**
> >
> > Thank you for the response. We need to better highlight the relevance of *batch* active anomaly detection and have added a paragraph to the related work section.
> >
> > Like Hoi et al., 2006a our work considers the batch active anomaly detection setting. Here it is assumed that labeling can be performed only once. Either because interacting with the expert for labeling is too expensive or because retraining the model with each additional query is too inefficient (Hoi et al., 2006a). The batch setting is widely studied in deep active learning (Sener and Savarese, 2018; Ash et al., 2020; Citovsky et al., 2021; Pinsler et al., 2019; Hoi et al., 2006b; Guo and Schuurmans, 2007) but is still under-explored in active anomaly detection.
> >
> > Hoi, Steven CH, et al. "Batch mode active learning and its application to medical image classification." Proceedings of the 23rd international conference on Machine learning. 2006.
> >
> > **Comparison to class-imbalanced classification.** An anomaly detection loss has a strong inductive bias that is missing in classification. While a classifier learns a decision boundary between two classes, an anomaly detector learns a decision boundary **around** a normal class. A semi-supervised anomaly detector (as we encounter in active anomaly detection) uses selected labeled anomalies to push the decision boundary tighter around the normal class. Notably, a purely imbalanced classifier would not detect unknown anomalies. In contrast, a semi-supervised anomaly detection method would still correctly label not previously encountered anomalies as abnormal.
> >
> > **Anomaly ratio estimation.** Note that estimating the contamination ratio can be beneficial for many anomaly detection algorithms, including OC-SVM (Schölkopf et al., 2001), kNN (Ramaswamy et al., 2000), Robust PCA/Auto-encoder (Zhou and Paffenroth, 2017), and Soft-boundary deep SVDD (Ruff et al., 2018). When working with contaminated data, these algorithms require a decent estimate of the contamination ratio for good performance.
> >
> > When estimating the anomaly ratios, the non-iid selected queried samples will form a different distribution from the original dataset, characterized by your SAT scores example, leading to a biased ratio estimation. To remove the bias, we proposed the importance-weighted estimator (Eq. 4), where the importance weights are useful to remove the selection bias. This technique requires two assumptions, and we empirically verified them. We also tested the robustness of the proposed unbiased estimator.

---

### Official Review · Reviewer_i6D3 · 2022-10-28

**Confidence:** 4
**Correctness:** 3
**Technical Novelty And Significance:** 3
**Empirical Novelty And Significance:** 2
**Recommendation:** 6

**Clarity, Quality, Novelty And Reproducibility:**

This is well-done work. Novelty is a bit limited since existing methods are combined. However, I am honouring the theoretical study found in the paper as a novelty. The paper is very well written and easy to follow. I see good reproducibility.

**Strength And Weaknesses:**

I like the rigorous analysis from a theoretical perspective and the extraordinary evaluation of their method on real data. The paper is well-written and builds upon existing publications. I do not agree with the assumption or application case. In my opinion, anomaly detection is no longer such a task if you are able to identify such anomalies from the training data. It then becomes a supervised task with the anomalous events being an additional class. This statement might sound philosophical. However, if you do active learning in an supervised setting, the authors ignore methods from active learning. One example is EMOC, see Freytag et al.: Selecting Influential Examples: Active Learning with Expected Model Output Changes. European Conference on Computer Vision (ECCV). If we know that there are novel classes in our training set (the known one is the inlier data, the novel one the outlier), we shall also study methods that combine active learning in a continuous learning scenario, for example, Kaeding et al.: Active and Continuous Exploration with Deep Neural Networks and Expected Model Output Changes. NIPS Workshop on Continual Learning and Deep Networks. 2016. I find that the authors should consider such work as well and include it in the comparison.

**Summary Of The Paper:**

The paper is a combination of Latent Outlier Exposure (Qiu 2022) and an active learning strategy from the same group. The idea is to exploit knowledge about the outlier rate to control active learning sample selection during training. The assumption is that finding anomalies in the (unlabeled) training set will improve the quality of anomaly detection in the test phase. A lot of experiments on different ML tasks (images, medical images, videos, etc.) demonstrate improvement over existing techniques. The authors also support their claims with some theorems.

**Summary Of The Review:**

The paper tackles a (from my perspective) very special (and not such common) case in anomaly detection. For me, it is not clear whether the detection of real anomalies and outliers can be improved. Real anomalies are those who have never been seen before, for example, novel classes in a continual learning scenario. It is quite natural that one can improve on anomaly detection if one can optimize the model during training on that already. Then, anomaly detection (even with only 10% outlier rate) boils down to a supervised setting (of course, an unbalanced one). Since the author missed to take this perspective as well and as a consequence to study and compare with literature in areas like normal active learning and continual learning, I am not that happy about the paper. Finally, I am missing experiments on image data beyond X-MNIST, like one can find results for normal active learning.

---

> ### Author Response · Authors · 2022-11-10
> **Response to Reviewer i6D3**
>
> We thank the reviewer for the overall positive assessment, particularly for the recognition of our theoretical contribution. We address the comments and suggestions below.
>
> > “I do not agree with the assumption or application case. In my opinion, anomaly detection is no longer such a task if you are able to identify such anomalies from the training data. It then becomes a supervised task with the anomalous events being an additional class... ”
>
> In anomaly detection, it is practical and realistic to assume that the unlabeled dataset is contaminated with unnoticed anomalies because ensuring the training set only contains clean normal data is difficult. For this reason, this setup has been studied widely in the community [Görnitz et al., 2013; Ruff et al., 2019; Qiu et al., 2022a]. In contrast to the supervised setting, the unlabeled dataset only contains a minor amount of anomalies that do not reveal the anomalous patterns completely. The scarcity of labeled anomalies makes the problem hard. Our proposed method outperformed previous work both in the proposed semi-supervised loss function and active querying strategies, which are demonstrated theoretically and empirically.
>
> > “If we know that there are novel classes in our training set (the known one is the inlier data, the novel one the outlier), we shall also study methods that combine active learning in a continuous learning scenario…”
>
> Continuous learning is a different problem setup and requires more human effort in the loop. In contrast, we only query the labels at once, causing minimal human effort in learning. See our response to reviewer uxaT’s comment 7.
>
> > “The paper tackles a (from my perspective) very special (and not such common) case in anomaly detection.”
>
> The setup we investigate is widely adopted in the literature. Please see our referenced baselines in the paper.
>
> > “For me, it is not clear whether the detection of real anomalies and outliers can be improved. Real anomalies are those who have never been seen before, for example, novel classes in a continual learning scenario.”
>
> Our work is not related to continual learning. We don't expect/assume to observe the complete characteristic features of anomalies, which is also infeasible in practice. However, querying a few anomalies provides valuable training signals, which has been studied in [Görnitz et al., 2013; Ruff et al., 2019; Qiu et al., 2022a]. With the outlier information, the generalization performance improves significantly [Ruff et al., 2019 Fig. 4; Qiu et al., 2022a Fig. 2
> ].
>
> > “Finally, I am missing experiments on image data beyond X-MNIST, like one can find results for normal active learning.”
>
> Our image datasets are standard anomaly detection benchmarks and they have been used virtually in all top-tier contributions in the field [Ruff et al., 2021]. We also included a diverse dataset MedMNIST, a recently released medical image benchmark. They are real, colorful medical images in tractable sizes. Besides, we evaluated the methods on 11 image datasets, which is much more than common works in this area.

---

### Author Response · Authors · 2022-11-10
**General comments**

We thank all reviewers for their valuable feedback, especially for finding our “theoretical study novel” and our paper “well-written and easy to follow.” Before replying to each reviewer individually, we would like to make the following general comments related to multiple reviews.

**The novelty of combining LOE and active learning strategies.** The combination of LOE and active learning is not an ad-hoc design. LOE benefits from active learning queries, and their relationship is characterized by Thm. 1. As follows, we demonstrate this relationship and stress our contribution.

The success of LOE relies on correctly ranking the training data according to their anomaly scores. However, the ranking is not guaranteed to be correct without additional information. The active queries and expert feedback provide a possibility of ensuring the ranking correctness of the training data. In Thm. 1, we theoretically build the relationship between ranking and queried data. The key is the radius of the cover generated by the queried samples -- the smaller the radius, the better the ranking correctness of the training data. We illustrate this relationship in Fig. 3 and Fig. 4, where we compute the radius by Eq. 5 and use AUROC to measure the correctness of the ranking. It shows the ranking correctness is negatively correlated with the cover radius, which verifies our theorem. Consequently, our proposed Kmeans++-based diversified querying strategy is a design choice derived from Thm. 1. Any other diversified querying strategy with a small cover radius can be another fit.

In addition, LOE relies on the assumed contamination ratio of the training data. Active learning provides an opportunity to estimate the contamination ratio through the queried ground truth labels. Towards that end, we derived an unbiased importance-weighted contamination ratio estimator with non-uniform queried data points and demonstrated the effectiveness empirically (in all our experiments and Table 4). We also believe this has a broader impact as the contamination ratio hyperparameter is widely used in established anomaly detection algorithms (see our response to reviewer uxaT’s comment 1).

**Comparison to supervised learning.** We update the model parameters with the labeled query data points (Eq. 1). Despite the similarity with supervised learning, anomaly detection faces a more challenging setup. Different from classification, novel classes of anomalies can occur during testing. An anomaly detector has to generalize to and correctly identify unseen anomalies. Even worse, anomalies are already scarce in the training data. Querying a few samples only allows the model to see minimal abnormal patterns. To resolve this difficulty, anomaly detection methods apply an inductive bias to the model, which leads to a compact representation of normal data. Any data located outside the compact normal data region is considered abnormal. On the other hand, supervised classifiers learn to tell apart known anomalies but fail to detect unseen ones. Ruff et al., 2019 illustrated this phenomenon in their paper Fig. 1 (c).

Ruff, Lukas, et al. "Deep Semi-Supervised Anomaly Detection." International Conference on Learning Representations. 2019.

---

> ### Author Response · Authors · 2022-11-19
> **Update of the Rebuttal Revision**
>
> We thank the reviewers for their thorough and valuable reviews. We considered all the comments and suggestions for preparing our revised paper. We clarified our work is for batch mode active anomaly detection and added related work of batch active learning; we stressed our theory applies to the optimization procedure of our proposed loss function. We also added time complexity analysis and the importance of estimating anomaly ratios.
>
> We thank all the reviewers' time and efforts again.
>
> Best wishes,
> Authors

---

### Decision · Program_Chairs · 2023-01-20

**Decision:**

Reject

**Justification For Why Not Higher Score:**

N/A

**Justification For Why Not Lower Score:**

N/A

**Metareview: Summary, Strengths And Weaknesses:**

This paper considers the problem of anamoly detection in an active learning setting where a batch of diverse queries are selected. The key idea is to select diverse cluster centers using k-means++ algorithm.

Clustering based active learning is a well-known idea in the classification setup. Some prototypical example references are:
https://www.cs.columbia.edu/~djhsu/papers/hier.pdf
https://icml.cc/Conferences/2004/proceedings/papers/94.pdf
and followup references.
A similar idea has also been explored by Das et al. as pointed out by one reviwer in the context of active anomaly detection using tree-based ensembles (e.g., Isolation Forest).
So the main idea lacks some novelty.

The proposed method relies on some strong and may not be practical assumptions. Similarly, experimental evaluation is lacking in many ways.
- Contamination ratio should be known/estimated. The authors' argue that their method is robust in estimating this quantity, but all experiments are run with a (artificial) fixed 10% contamination factor. This claim requires more evidence with more comprehensive experiments.
- Empirical evaluation is done using a batch of queries selected in a single round. The paper needs to argue the motivation and real-world applications behind this problem setting. Otherwise, there is a lot of recent work on human-in-the-loop learning for anomaly detection which queries in several rounds. Even if the setting considered is practical, it would have been interesting to see how the proposed method performs in several rounds of batch queries.
- Some of the assumptions need more clarification in the main paper and possibly a good intuitive illustration. Since the method relies on these assumptions, it is important to convince the reader about them.
- More experiments on realistic anomaly detection tasks where the fraction of anomalies are less than two percent will help understand the overall effectiveness of the proposed method.

The paper is promising, but it falls short of acceptance for the above reasons. Hence, I recommend rejecting the paper. I strongly encourage the authors' to improve the paper based on the review feedback for resubmission.